EMBO
Molecular Medicine

# *KLB*, encoding β-Klotho, is mutated in patients with congenital hypogonadotropic hypogonadism

Cheng Xu[1], Andrea Messina[1], Emmanuel Somm[1], Hichem Miraoui[1,†], Tarja Kinnunen[2], James Acierno Jr[1], Nicolas J Niederländer[1], Justine Bouilly[1], Andrew A Dwyer[1,3], Yisrael Sidis[1], Daniele Cassatella[1], Gerasimos P Sykiotis[1], Richard Quinton[4], Christian De Geyter[5], Mirjam Dirlewanger[6], Valérie Schwitzgebel[6], Trevor R Cole[7], Andrew A Toogood[8], Jeremy MW Kirk[9], Lacey Plummer[10], Urs Albrecht[11] , William F Crowley Jr[10], Moosa Mohammadi[12], Manuel Tena-Sempere[13,14,15], Vincent Prevot[16,17] & Nelly Pitteloud[1,*]

## Abstract

Congenital hypogonadotropic hypogonadism (CHH) is a rare genetic form of isolated gonadotropin-releasing hormone (GnRH) deficiency caused by mutations in > 30 genes. Fibroblast growth factor receptor 1 (*FGFR1*) is the most frequently mutated gene in CHH and is implicated in GnRH neuron development and maintenance. We note that a CHH *FGFR1* mutation (p.L342S) decreases signaling of the metabolic regulator FGF21 by impairing the association of FGFR1 with β-Klotho (KLB), the obligate co-receptor for FGF21. We thus hypothesized that the metabolic FGF21/KLB/FGFR1 pathway is involved in CHH. Genetic screening of 334 CHH patients identified seven heterozygous loss-of-function *KLB* mutations in 13 patients (4%). Most patients with *KLB* mutations (9/13) exhibited metabolic defects. In mice, lack of *Klb* led to delayed puberty, altered estrous cyclicity, and subfertility due to a hypothalamic defect associated with inability of GnRH neurons to release GnRH in response to FGF21. Peripheral FGF21 administration could indeed reach GnRH neurons through circumventricular organs in the hypothalamus. We conclude that FGF21/KLB/FGFR1 signaling plays an essential role in GnRH biology, potentially linking metabolism with reproduction.

**Keywords** beta-klotho; congenital hypogonadotropic hypogonadism; fibroblast growth factor 21; fibroblast growth factor receptor 1
**Subject Categories** Genetics, Gene Therapy & Genetic Disease; Metabolism

See also: **M Misrahi** (October 2017)

## Introduction

Congenital hypogonadotropic hypogonadism (CHH) is a disorder characterized by absent or partial puberty and infertility due to isolated gonadotropin-releasing hormone (GnRH) deficiency. When CHH is associated with anosmia, the term Kallmann syndrome (KS) is used. The genetics of CHH is complex with mutations in over 30 genes known to be involved, and with both classical and complex modes of inheritance, such as oligogenicity (Boehm *et al*, 2015; Stamou *et al*, 2016).

Fibroblast growth factor receptor 1 (FGFR1) is essential for cell proliferation, differentiation, and migration during embryonic

---

1 Service of Endocrinology, Diabetology & Metabolism, Lausanne University Hospital, Lausanne, Switzerland
2 Department of Biology, School of Applied Sciences, University of Huddersfield, Huddersfield, UK
3 University of Lausanne Institute of Higher Education and Research in Healthcare, Lausanne, Switzerland
4 Institute for Genetic Medicine, University of Newcastle-on-Tyne, Newcastle-on Tyne, UK
5 Clinic of Gynecological Endocrinology and Reproductive Medicine, University Hospital, University of Basel, Basel, Switzerland
6 Pediatric Endocrine and Diabetes Unit, Children's Hospital, University Hospitals and Faculty of Medicine, Geneva, Switzerland
7 Department of Clinical Genetics, Birmingham Women's Hospital, Birmingham, UK
8 Department of Endocrinology, Queen Elizabeth Hospital, University Hospitals Birmingham, Birmingham, UK
9 Department of Endocrinology, Birmingham Children's Hospital, Birmingham, UK
10 National Center for Translational Research in Reproduction and Infertility, Harvard Reproductive Endocrine Sciences Center of the Department of Medicine, Massachusetts General Hospital, Boston, MA, USA
11 Department of Biology, Biochemistry, Faculty of Science, University of Fribourg, Fribourg, Switzerland
12 Department of Biochemistry & Molecular Pharmacology, New York University School of Medicine, New York, NY, USA
13 Department of Cell Biology, Physiology and Immunology, University of Cordoba, Cordoba, Spain
14 Instituto Maimonides de Investigación Biomédica de Cordoba (IMIBIC/HURS), Cordoba, Spain
15 CIBER Fisiopatología de la Obesidad y Nutrición, Instituto de Salud Carlos III, Cordoba, Spain
16 Inserm, Laboratory of Development and Plasticity of the Neuroendocrine Brain, JPARC, Lille, France
17 FHU 1000 Days for Health, School of Medicine, University of Lille, Lille, France
 *Corresponding author. Tel: +41 021 314 87 96; E-mail: nelly.pitteloud@chuv.ch
 †Present address: Division of Genetics, Department of Medicine, Brigham and Women's Hospital, Harvard Medical School, Boston, MA, USA

---

development. Gain-of-function mutations in *FGFR1* cause several skeletal disorders such as Pfeiffer syndrome (MIM: 101600) and Jackson–Weiss syndrome (MIM: 123150). In 2003, *FGFR1* was identified as the first gene underlying the autosomal dominant form of KS (Dode *et al*, 2003), and subsequently, loss-of-function *FGFR1* mutations were identified in CHH patients with normal olfaction (Pitteloud *et al*, 2006). Altogether, *FGFR1* mutations are present in approximately 10% of CHH cases and are often associated with incomplete penetrance and variable expressivity (Miraoui *et al*, 2011). Non-reproductive defects associated with *FGFR1* mutations in CHH patients include anosmia, cleft lip/palate, dental agenesis and split-hand/foot malformation (Costa-Barbosa *et al*, 2013; Villanueva *et al*, 2015).

The identification of a KS patient carrying an *FGFR1* p.L342S mutation was informative in identifying FGF as a critical ligand of FGFR1 in GnRH biology and in documenting *FGF8* as a gene mutated in CHH. The FGFR1 L342S mutant selectively disrupts FGF8 signaling leaving FGF1 or FGF2 signaling unaffected (Pitteloud *et al*, 2007). This observation led to the identification of FGF8 as a critical morphogen for GnRH neuron ontogenesis (Pitteloud *et al*, 2007; Chung *et al*, 2008; Falardeau *et al*, 2008). Subsequently, several other mutations were identified in genes within the *FGF8* genetic network (e.g. *IL17RD* and *FGF17*), thus reinforcing the importance of the FGF8/FGFR1 pathway in CHH pathogenesis (Miraoui *et al*, 2013).

Murine studies have documented and characterized the role of FGFR1, not only in GnRH neuron fate specification, but also in GnRH network homeostasis. Mice with dominant-negative FGF receptor (dnFGFR) targeted to GnRH neurons exhibit delayed puberty, early reproductive senescence, and loss of > 50% of GnRH neurons in adulthood (Tsai *et al*, 2005). Furthermore, intracerebroventricular (ICV) injection of anti-FGFR1 antibodies induces rapid and significant weight loss, indicating a role of *Fgfr1* in the central regulation of metabolism (Sun *et al*, 2007). Because Fgf8 expression is mostly restricted to embryonic development (Fon Tacer *et al*, 2010), we hypothesized that a different ligand for FGFR1 might be implicated in postnatal GnRH biology. FGF21, an

endocrine FGF mainly secreted by the liver, has been identified as a major peripheral and central metabolic regulator (Owen *et al*, 2015). FGF21 binds with low affinity to heparan sulfate and requires β-Klotho as co-receptor to signal preferentially through its receptor, FGFR1 isoform c (FGFR1c; Goetz *et al*, 2007; Kurosu *et al*, 2007; Ogawa *et al*, 2007). Notably, β-Klotho competes with FGF8 for the same binding site in the immunoglobulin D3 domain of FGFR1c (Goetz *et al*, 2012). We thus hypothesized that defects in the FGF21/KLB/FGFR1 signaling pathway may underlie GnRH deficiency in both humans and rodents.

## Results

### The CHH-associated FGFR1 L342S mutation alters FGF21 signaling

The FGFR1 p.L342S mutation identified in a KS proband (Pitteloud *et al*, 2007) disrupted not only FGF8 signaling, but also FGF21 signaling (Fig 1A and B), consistent with the same binding pocket on FGFR1c competed between FGF8 and β-Klotho (Goetz *et al*, 2012). These data led to the hypothesis that FGF21 signaling through FGFR1c and β-Klotho might be important for GnRH neuron biology. Interestingly, the KS patient harboring the p.L342S mutation also exhibits metabolic phenotypes: he is obese (BMI of 30 kg/m$^2$) with increased fat mass (40% by DEXA, > 97$^{th}$ percentile; Kelly *et al*, 2009), and severe insulin resistance (evaluated by hyperinsulinemic euglycemic clamp study, low glucose disposal rate at 1.5 mg/kg/min; Bergman *et al*, 1985).

### CHH patients harbor putative pathogenic variants in *KLB*, encoding the FGF21 co-receptor β-Klotho

Targeted sequencing in an unselected cohort of CHH patients ($n$ = 334) revealed no rare sequencing variants (RSVs) in *FGF21*. In contrast, seven heterozygous putative pathogenic RSVs in *KLB* were identified among 13 CHH probands: p.R309W, p.R309Q, p.R424C,

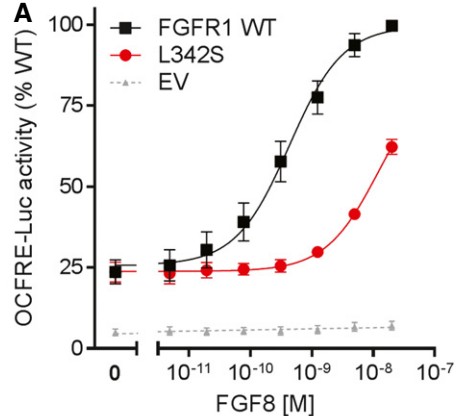
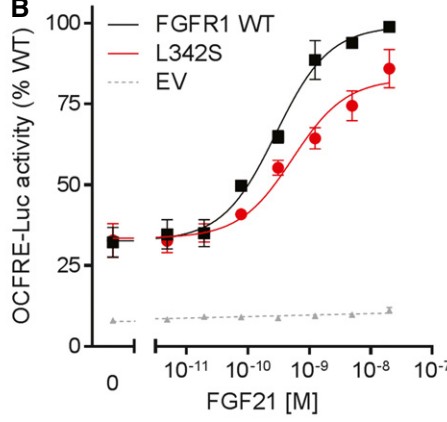

**Figure 1.  FGFR1 L342S luciferase assays.**

A, B    Reporter activity of FGFR1 WT and L342S stimulated with increasing doses of FGF8 (A) and FGF21 (B). FGFR1 L342S severely affects FGF8 signaling, with a 28-fold increase in EC$_{50}$ ($P$ < 0.001). L342S also impairs FGF21 signaling, with a 20% reduction in maximal response ($P$ = 0.001). The experiments were repeated three times.  Data were plotted as mean ± SEM. Maximal responses and EC$_{50}$ of mutants were compared to WT by *F*-test. WT, wild type; EV, empty vector.

p.A574T, p.F777delF, p.K815E, p.L1011P (Fig EV1A, Table 1). None of these variants were predicted to affect splicing. The inframe deletion (p.F777delF) was identified in seven unrelated patients of European ancestry and was not seen in our in-house reproductively normal controls ($n = 191$). Recurrent pathogenic variants are uncommon in CHH, except for a few founder mutations such as *PROKR2* p.L173R, *GNRH1* c.18-19insA, and *GNRHR* p.R139H (Bouligand *et al*, 2009; Avbelj Stefanija *et al*, 2012; Beneduzzi *et al*, 2014; Choi *et al*, 2015). Haplotype analysis based on sequence variants within *KLB* was insufficiently informative to distinguish between a founder effect and a mutational hot spot for p.F777delF. This variant is significantly more frequent in the CHH cohort compared to ethnically matched controls from ExAC database ($P = 0.01$), as are the other *KLB* mutations except p.R309W and p.K815E (Table 1). In addition, none of the identified *KLB* variants were found in a homozygous status in the ExAC database.

### *KLB* variants have impaired functionality *in vitro*

The functional effect of the identified *KLB* variants on FGF21 signaling was tested via cell-based reporter gene assay (Raivio *et al*, 2009). L6 myoblast cells expressing wild-type (WT) FGFR1c and KLB showed a typical sigmoid dose–response curve and elicited a fourfold increase in the activity of the osteocalcin reporter when treated with increasing doses of rFGF21 (0–20 nM; Fig 2A and B). All mutants, except R309W, exhibited significantly decreased maximal response to FGF21 compared to WT ($P < 0.05$; Fig 2A and B); R309W exhibited a threefold increase in $EC_{50}$ compared to WT (0.52 M vs. 0.15 M, $P < 0.001$), suggesting a decreased affinity for FGF21.

Through biochemical studies, we have previously shown that KLB enhances FGF21-FGFR1c binding and hence promotes FGF21 signaling by simultaneously tethering FGF21 and FGFR1c to itself through two distinct sites (Goetz *et al*, 2012). To test whether the reduced FGF21 signaling was due to a significant defect in the receptor complex formation, we tested the interaction between KLB mutants and FGFR1c through a co-immunoprecipitation (co-IP)

assay. All of the KLB mutants co-precipitated with FGFR1c to the same extent as the wild-type KLB (Fig 2C), indicating that the KLB mutants do not exhibit a major defect in the binding to FGFR1.

Overall expression levels of the KLB mutants were similar to WT except for R309W which was slightly decreased vs. WT (Fig EV1B). However, the mature KLB fractions of R309W, R424C, K815E, and L1011P were decreased (Fig EV1B), suggesting defects in protein glycosylation. The R309W, R309Q, F777delF, K815E, and L1011P mutants exhibited significantly less cell surface expression compared to WT (Fig 2D), consistent with their impaired signaling (Fig 2A and B). Collectively, our *in vitro* studies showed decreased function of the KLB mutants either through reduction in signaling, ligand affinity, or expression. Furthermore, the KLB F777delF and FGFR1 R78C mutants co-occurring in Subject 9 (Table 2) were further tested using the same FGF reporter assay to model the effect of digenicity in the FGFR1 pathway. While the individual KLB F777delF or FGFR1 R78C mutants each evoked a decreased response (15 and 20%, respectively, $P < 0.05$), the pairing of the co-receptor/receptor mutants showed further decreased activity (35%), confirming an additive effect of the digenic mutation ($P < 0.05$, Fig 2E).

### KLB mutants fail to rescue KLB homolog function in *Caenorhabditis elegans*

To investigate the functional effect of *KLB* mutations *in vivo*, we performed rescue assays in *C. elegans* (Neumann-Haefelin *et al*, 2010). The role of FGFR and klotho proteins as regulators of metabolic homeostasis is evolutionarily conserved in the nematode *C. elegans* (Kokel *et al*, 1998; Polanska *et al*, 2011). *Caenorhabditis elegans* has a single *FGFR* homolog, *egl-15* (DeVore *et al*, 1995) and two Klotho/KLB homologs, *klo-1* and *klo-2* (Polanska *et al*, 2011). Double mutants of *klo-1* (ok2925) and *klo-2* (ok1862) are viable but subfertile with a reduction of ~40% in egg laying compared to WT. In addition, these worms display a fluid homeostasis phenotype, manifesting as the appearance of hollow cysts within the body cavity in 31% of animals ($n = 174$, Fig 2F and G); this phenotype is

**Table 1. Summary of *KLB* mutations identified in patients with congenital hypogonadotropic hypogonadism.**

| KLB mutations | Nucleotide change | Number of subjects | | MAF ExAC | P-value | SIFT/ PPH2 | *In vitro* studies | | | | Rescue assay in *Caenorhabditis elegans* | Interpretation ACMG guidelines |
| | | CHH ($n = 334$) | Control ($n = 191$) | | | | FGF21 signaling | Cell surface expression | Overall expression | Maturation level | | |
|---|---|---|---|---|---|---|---|---|---|---|---|---|
| p.R309W | c.925C>T | 1 | 0 | $8.9e10^{-5}$ | 0.07 | 2/2 | ↓ | ↓ | ↓ | ↓ | None | Likely pathogenic |
| p.R309Q | c.926G>A | 1 | 0 | 0 | 0.01 | 1/2 | ↓ | ↓ | NS | NS | Partial | Likely pathogenic |
| p.R424C | c.1270C>T | 1 | 0 | 0 | 0.01 | 1/2 | ↓ | NS | NS | ↓ | Partial | Likely pathogenic |
| p.A574T | c.1720G>A | 1 | 0 | 0 | 0.04 | 1/2 | ↓ | ↑ | NS | NS | Partial | Likely pathogenic |
| p.F777delF | c.2329delTTC | 7 | 0 | 0.004 | 0.01 | NA | ↓ | ↓ | NS | NS | None | Pathogenic |
| p.K815E | c.2443A>G | 1 | 1 | 0.001 | 0.57 | 2/2 | ↓ | ↓ | NS | ↓ | Partial | Likely pathogenic |
| p.L1011P | c.3032T>C | 1 | 0 | $6.0e10^{-5}$ | 0.049 | 2/2 | ↓ | ↓ | NS | ↓ | None | Uncertain significance |

CHH, congenital hypogonadotropic hypogonadism; MAF, minor allele frequency; *P*-value, Fisher's exact test of allele frequency in CHH cohort versus ethnically matched controls from ExAC database; PPH2, polyphen-2; ↓, decreased; ↑, increased; NS, not significant. ACMG, American College of Medical Genetics and Genomics (Richards *et al*, 2015).

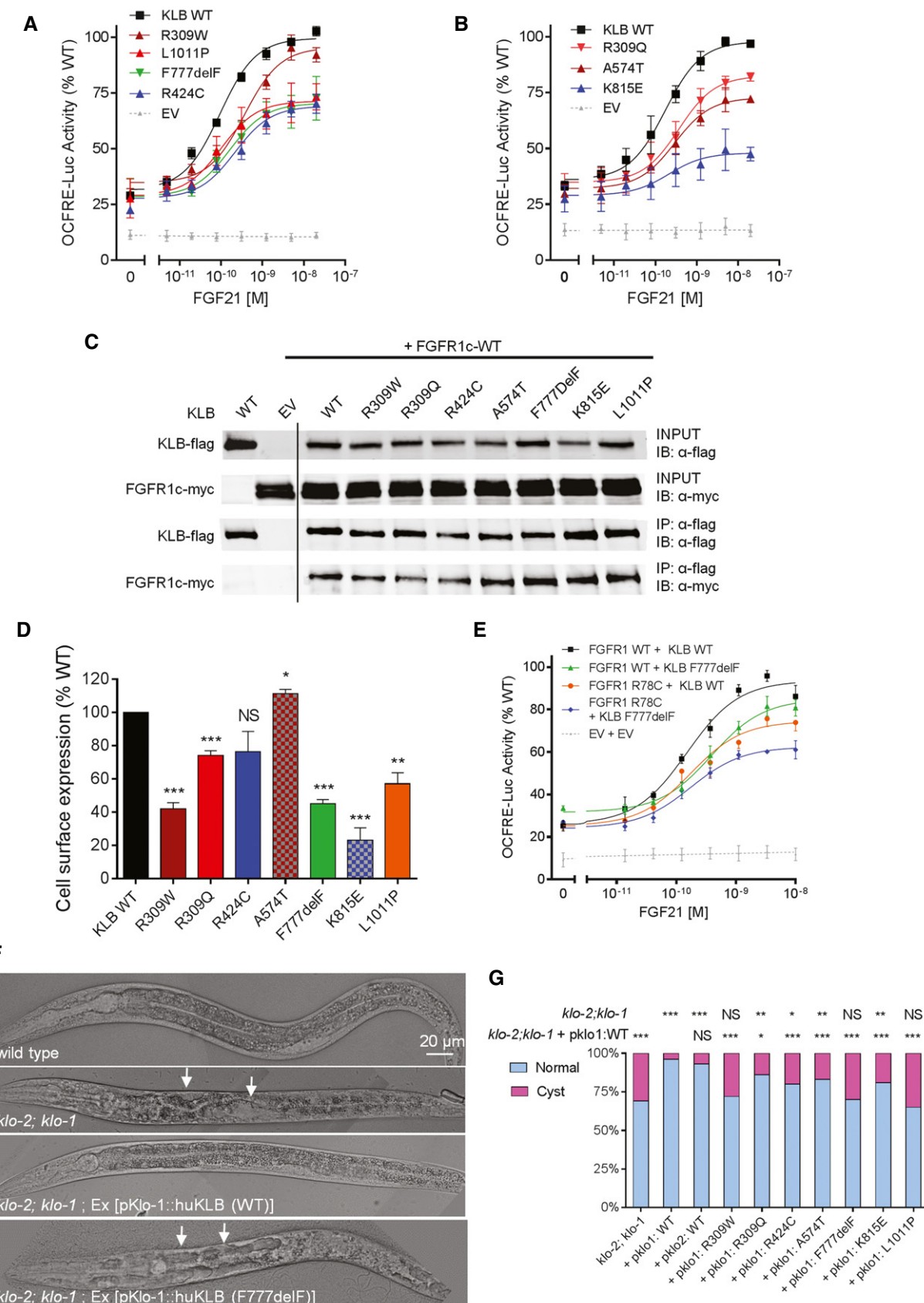

Figure 2.

◄

**Figure 2.  KLB functional assays.**

A, B   Reporter activity of KLB WT and mutants stimulated with FGF21. The maximal response to FGF21 is impaired ($P < 0.05$) in all but one KLB mutant, R309W, which exhibits nevertheless increased $EC_{50}$ compared to WT ($P < 0.001$). All luciferase activity assays were plotted as mean $\pm$ SEM of three independent experiments. Maximal responses and $EC_{50}$ of mutants were compared to wild type (WT) by *F*-test.

C   Immunoprecipitation (IP) assay. HEK293T cells were transfected with flag-tagged KLB and myc-tagged FGFR1c. KLB was immunoprecipitated with monoclonal anti-flag antibody. KLB was detected with anti-flag antibody and FGFR1c with anti-myc antibody. IB: immunoblotting. Experiments were repeated twice with similar results. Lanes presented in the figure are from different blots and were juxtaposed.

D   Cell surface expression of KLB WT and mutants. Data are presented as the mean $\pm$ SEM of three independent experiments; mutants were compared to WT by unpaired *t*-test.

E   Activity of the double mutant FGFR1 R78C and KLB F777delF. The double mutant exhibits a further decrease in response to FGF21 compared to either single mutant ($P < 0.05$). The experiments were repeated three times, and data were plotted as mean $\pm$ SEM. Maximal responses of mutants were compared to WT by *F*-test.

F   Representative DIC images of rescue assays in *Caenorhabditis elegans*. Worms deficient in both *klo-1* and *klo-2* develop hollow cysts within their body cavity (panel 2, arrows). Transgenic expression of WT human KLB in *klo-1; klo-2* mutant worms rescue this phenotype (panel 3). Transgenic expression of the KLB F777delF mutant fails to rescue the phenotype (panel 4, arrows).

G   Quantification of the rescue assay results. *klo-1; klo-2* double mutant worms were injected with human KLB WT and mutant constructs under the control of pklo1 or pklo2 promoters. Each bar represents the average of two to four independent transgenic lines. For each KLB mutant, the percentage of worms with cystic phenotypes was compared to the *klo-1; klo-2* double mutant and to the pklo1:WT controls by Fisher's exact test.

Data information: WT, wild type; EV, empty vector. *$P < 0.05$, **$P < 0.01$, ***$P < 0.001$, NS: not significant.
Source data are available online for this figure.

**Table 2.  Clinical phenotypes of congenital hypogonadotropic hypogonadism patients with heterozygous *KLB* mutations.**

| Subject | *KLB* mutation | Sex | Age at Dx | Inheritance | Puberty | Olfaction | Additional reproductive phenotypes | BMI (kg/m²) | Metabolic phenotypes | Other CHH gene mutations |
|---|---|---|---|---|---|---|---|---|---|---|
| 1[a] | p.R309W | M | 25 | F | A | A | Cryptorchidism | 40 | Obesity, insulin resistance | None |
| 2 | p.R309Q | M | 28 | F | P | A | None | 25 | Impaired fasting glucose | *FGF8* p.P26L |
| 3 | p.R424C | M | 31 | F | A | A | Cryptorchidism | 28 | Overweight | None |
| 4 | p.A574T | M | 19 | F | A | N | Micropenis | 22 | Impaired fasting glucose, dyslipidemia | *PROKR2* p.S188L |
| 5 | p.F777delF | F | 18 | F | A | N | None | 23 | NA | *GNRHR* p.Q106R |
| 6 | p.F777delF | F | 16 | F | A | H | None | 17 | Underweight | None |
| 7 | p.F777delF | M | 18 | F | A | A | Cryptorchidism, micropenis | 43 | Obesity, insulin resistance, dyslipidemia | None |
| 8 | p.F777delF | M | 19 | S | P | A | Reversal | 21 | Dyslipidemia | None |
| 9 | p.F777delF | M | 16 | S | A | A | Micropenis, retractile testes | 25 | Overweight, dyslipidemia | *FGFR1* p.R78C |
| 10 | p.F777delF | M | 25 | S | A | A | Cryptorchidism | 26 | Overweight | None |
| 11[a] | p.F777delF | M | 16 | S | A | A | Hypospadias, cryptorchidism | 20 | Insulin resistance | *PROKR2* p.L173R |
| 12 | p.K815E | M | 53 | S | P | A | Fertile eunuch | 22 | None | None |
| 13 | p.L1011P | M | 19 | F | P | A | Micropenis | 20 | NA | None |

Sex: F, female; M, male; Dx, diagnosis; Inheritance: F, familial; S, sporadic; Puberty: A, absent; P, partial; Olfaction: A, anosmia; N, normosmia; H, hyposmia; NA, not available.
[a]Subjects from the exome cohort..

presumably related to the role of *klo-1* in excretory canal development in *C. elegans* (Polanska *et al*, 2011). Transgenic expression of wild-type human KLB under the control of 5′ upstream sequences (referred to as "promoter") of either the *C. elegans klo-1* or *klo-2* was able to rescue the worm *klo-2; klo-1* double mutant phenotype (Fig 2F and G). In contrast, transgenic expression of human KLB containing F777delF, R309W, or L1011P mutations under the control of the *klo-1* promoter failed to rescue the cyst phenotype

(Fig 2G). All other mutants displayed decreased rescue ability (Fig 2G).

In summary, all *KLB* variants have impaired functionality *in vitro* and *in vivo*, and can thus be considered mutations (Table 1). Integrating evidence from population data, computational algorithms, functional assays, and segregation data according to the guidelines of the American College of Medical Genetics and Genomics (ACMG; Richards *et al*, 2015; Li & Wang, 2017), all *KLB* variants except p.L1011P were classified as "pathogenic" or "probably pathogenic", while p.L1011P as "variant of uncertain significance" (Table 1).

### Genotype–Phenotype correlations

The clinical data of the 11 male and two female CHH patients with *KLB* mutations are summarized in Table 2 and Fig 3. Five cases were sporadic while the remainder (8/13, 62%) exhibited a familial inheritance, which is twice the known overall frequency of familial cases in CHH (30%; Raivio *et al*, 2009).

The majority of patients (7 males, 2 females) presented with absent puberty and micropenis and/or cryptorchidism in males—phenotypes consistent with severe GnRH deficiency; four patients exhibited partial puberty, indicating less severe GnRH deficiency. One patient underwent spontaneous reversal after discontinuing testosterone treatment (Patient 8; Raivio *et al*, 2007). The majority of patients (11/13) had abnormal olfaction and were thus diagnosed as KS. In addition, nine of 13 of these young CHH patients had metabolic phenotypes, including overweight/obesity, dyslipidemia, and/ or insulin resistance; one patient was underweight at diagnosis (Table 2). Among the seven patients carrying the p.F777delF mutation, only one had partial puberty and reversal of HH. This patient carries only the *KLB* mutation (Table 2). Additional CHH-associated phenotypes were highly variable with polydactyly, renal phenotypes, synkinesia, or cleft lip and palate present in one patient each (Appendix Case Reports).

DNA from family members was available for 10 pedigrees, none of which revealed *de novo* mutations. Variable expressivity and

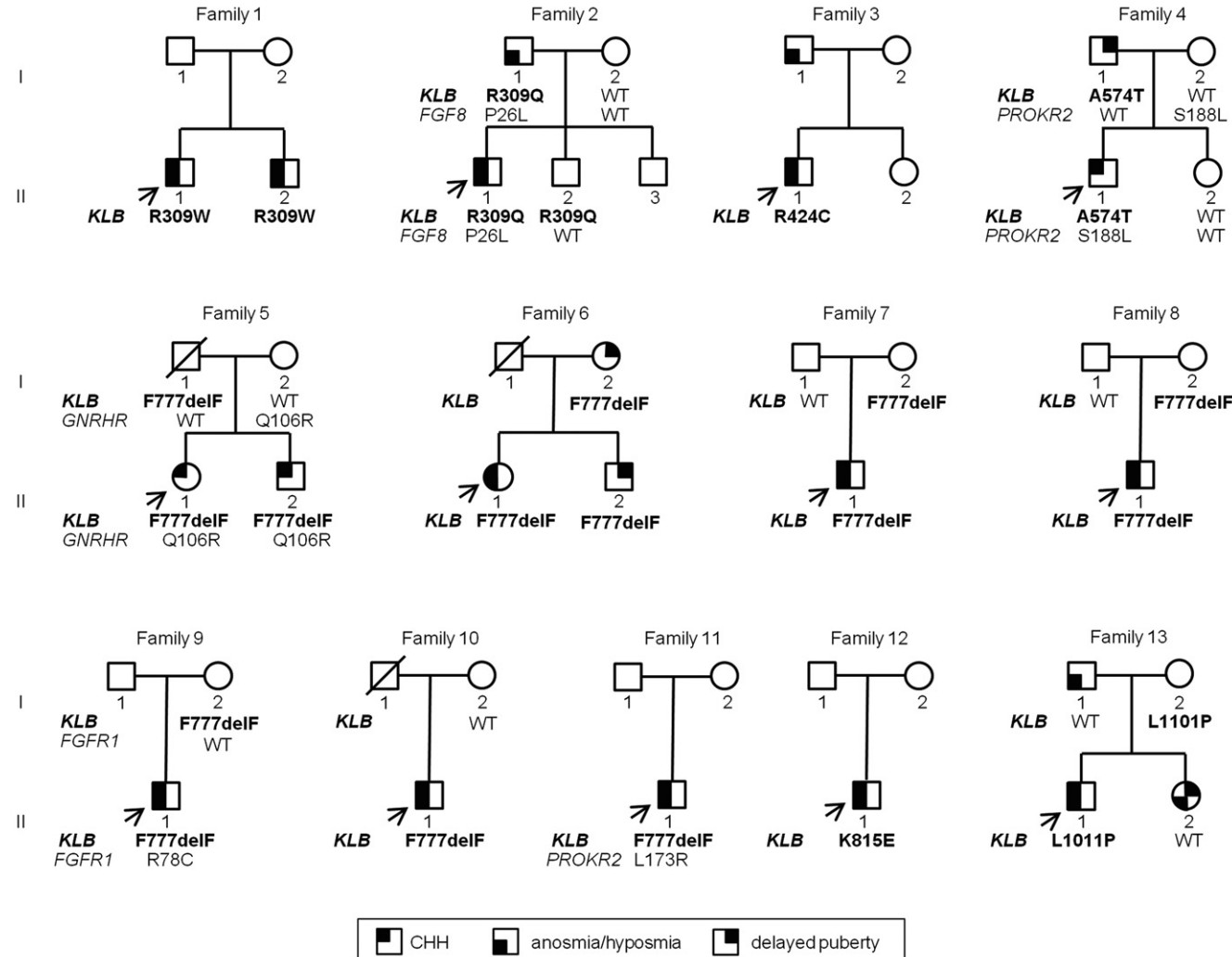

**Figure 3.  Pedigrees of CHH probands harboring *KLB* mutations.**

All identified *KLB* mutations are heterozygous. CHH, congenital hypogonadotropic hypogonadism; circles denote females; squares denote males; arrows depict probands; WT denotes wild type.

incomplete penetrance, common features in CHH, were observed in families harboring *KLB* mutations: family members in pedigrees 2, 4, and 6 carry the same *KLB* mutation as the proband yet exhibit attenuated phenotypes (delayed puberty or isolated anosmia), indicating variable expressivity. Asymptomatic carriers are present in families 2, 5, 7, 8, 9, and 13. Five probands (35%) were found to carry another heterozygous mutation in a known CHH gene in addition to the *KLB* mutation (Fig 3, Table 2), comprising *FGF8* (Subject 2), *PROKR2* (Subjects 4 and 11), *GNRHR* (Subject 5), and *FGFR1* (Subject 9). Additional mutations were not observed in our in-house control subjects ($n = 191$). The additional mutations are predicted to be deleterious *in silico* and four of five have been previously shown to be loss of function *in vitro* (Appendix Case Reports). Within the families with mutations in two different genes (Pedigree 4 and 5), we observed a more severe phenotype in the probands with digenic mutations compared to their parents with single gene defects. Further, among the four probands with partial GnRH deficiency, three patients carry heterozygous *KLB* mutation alone. These observations suggest that some degree of oligogenicity is present in this population and that the genetic load may impact the phenotypic severity. Metabolic data were not available for family members, and therefore, we are unable to assess these features in family members carrying *KLB* mutations.

Considering the broader FGF21/KLB/FGFR1 gene network in this CHH cohort, 38 *FGFR1* mutations were identified (including a homozygous one) in 45 of 334 (13%) of patients (Appendix Table S1); these include the patient (Subject 9) carrying heterozygous mutations in *KLB* and *FGFR1* mentioned above (Table 2). Thus, in total, 57 of 334 (17%) of CHH patients harbor mutation(s) in *FGF21/KLB/FGFR1*, either as monoallelic mutations or as digenic combinations.

### Klb-deficient mice exhibit delayed puberty and subfertility due to a hypothalamic defect

To further establish the role of *Klb* in GnRH biology, we studied the reproductive phenotype of *Klb* knockout mice (KlbKO). In the C57BL/6J pure background, we observed a high neonatal mortality of KlbKO mice (KlbKO represented 5–10% of the litters versus the 25% expected from heterozygous breeding pairs). The KlbKO mice exhibited marked delay of vaginal opening and first estrus (Fig 4A and B), measures of the initiation and completion of puberty, respectively. Notably, KlbKO mice displayed reduced growth, with a lower weight that persists in adulthood (Fig 4C and D). However, the reduced weight alone did not explain the delayed puberty, as KlbKO females reached puberty at a significantly lower weight than WT (Fig EV2A and B). Uterine weight, body composition, food intake, energy expenditure, and glucose tolerance were similar to WT (Figs 4E and EV2C–J). Serum insulin, leptin, and cholesterol levels were normal (Fig EV2K–L).

In accordance with their reduced ovarian weight (normalized to total body weight) and their decreased number of corpora lutea (Fig 4F–I), KlbKO adult females of 3–4 months old had irregular estrous cycles compared to their WT counterparts, with a significant decrease of the time spent in estrous (Fig 4J–K). In addition, KlbKO mice displayed reduced fertility compared to WT mice, yielding significantly fewer litters in the continuous mating protocol fertility test breeding with WT males (Fig 4L).

The exposure of WT female mice to sexually mature male pheromones leads to the initiation of proestrus, including increased hypothalamic secretion of GnRH and a preovulatory LH surge (Bronson & Stetson, 1973). In contrast, KlbKO females exhibit a substantially blunted LH surge compared to WT (Fig 4M). Intraperitoneal (IP) GnRH injections elicited a robust rise in LH in both WT and KlbKO mice, excluding a pituitary defect (Fig 4N). Furthermore, IP Kisspeptin (Kp-10) injections also induced LH secretion in both in WT and KlbKO females, suggesting that GnRH neurons are present and can respond to stimulation (Fig 4O). These results illustrate an implication of *Klb* in hypothalamic GnRH secretion, consistent with a contribution of *Klb* to the central regulation of reproduction.

As *KLB* mutations are heterozygous in CHH patients, we further tested whether *Klb* heterozygous (KlbHET) female mice display a reproductive phenotype. We observed that KlbHET exhibited similar reproductive defects as KlbKO mice: disrupted estrous cycle (Fig EV3A and B) and blunted LH surge at the estrus stage (Fig EV3C–F). In addition, the short-term fertility test showed a reduced pregnancy rate in KlbHET females, indicating a fertility

---

**Figure 4.   Lack of *Klb* in female mice leads to defects in sexual maturation with delay of puberty, altered cyclicity, reduced LH secretion, and reduced fertility.**

A–C   Altered pubertal development in KlbKO mice compared to WT littermates including delayed vaginal opening (KO: $n = 12$; WT: $n = 11$) and first estrous (KO: $n = 7$; WT: $n = 11$), expressed as cumulative percentage as a function of age (Gehan–Breslow–Wilcoxon test), and reduced body weight (KO: $n = 7$; WT: $n = 7$; two-way RM ANOVA).

D   Body weight of adult KlbKO mice ($n = 10$) compared to WT littermates ($n = 15$), using unpaired *t*-test.

E   Body composition expressed as percentage of fat mass in KlbKO ($n = 20$) and WT mice ($n = 23$), unpaired *t*-test.

F   Representative low magnification picture of the uterus and ovaries in WT and KlbKO mice. Scale bar 1 mm.

G   Hematoxylin–eosin-stained ovarian representative sections of KlbKO and WT mice at 3 months (CL: corpora lutea). Scale bar: 600 μm.

H   Normalized ovarian weight expressed as ovary/body weight ratio in KlbKO ($n = 7$) and WT mice ($n = 10$), unpaired *t*-test.

I   Quantification of corpora lutea per ovary in KlbKO ($n = 5$) and WT ($n = 5$) adult females, unpaired *t*-test.

J   Representative estrous cycle patterns of 3- to 4-month-old WT and mutant females, demonstrating marked alteration in KlbKO.

K   Quantification of time spent in different estrous cycle phases in KlbKO ($n = 13$) and WT ($n = 17$) adult females, unpaired *t*-test.

L   Fertility evaluated as number of litters per female and litter size in KlbKO ($n = 6$) and WT littermates ($n = 5$) in a continuous mating protocol (7 months), unpaired *t*-test.

M   KlbKO females display blunted preovulatory LH surge in male-induced proestrous (MIP) test (WT Di: $n = 5$; WT Pro: $n = 8$; KO Di: $n = 5$; KO Pro: $n = 5$).

N   GnRH test: basal blood LH levels and 30 min after intraperitoneal GnRH injection (0.25 μg) show LH release in both KlbKO ($n = 5$) and WT littermates ($n = 7$).

O   Kisspeptin test: blood LH levels at basal state and 30 min after intraperitoneal Kisspeptin injection (Kp-10; 1 nmole) in WT ($n = 5$) and KlbKO ($n = 5$) female mice.

Data information: Multiple comparison analysis of LH levels was performed using a two-way ANOVA followed by Fisher's LSD test (for MIP) and Sidak's multiple comparison test (for GnRH and Kisspeptin tests). Values shown are mean ± SEM; *$P < 0.05$, **$P < 0.01$, ***$P < 0.001$. ns: not significant.

defect (Fig EV3G). These findings indicate that haploinsufficiency for *Klb* leads to significant reproductive defects in mice.

### *Klb* is implicated in GnRH biology in the postnatal period but not during embryonic development

To further explore the defect of GnRH secretion in KlbKO mice, we first localized and quantified the GnRH neuron population in adult brains (P90–180). A normal complement of hypothalamic GnRH neurons and axonal projections to the median eminence (ME) was found in KlbKO mice (Fig 5A–C), indicating normal embryonic GnRH neurons development. We next investigated the hypothalamic expression of *Gnrh* and other genes known to be important for puberty and GnRH biology (i.e., *Kiss1*, *Npy*, *Pomc*, and *Lepr*); expression of these genes was not altered in KlbKO mice (Fig EV4A and B). Finally, we found no alteration

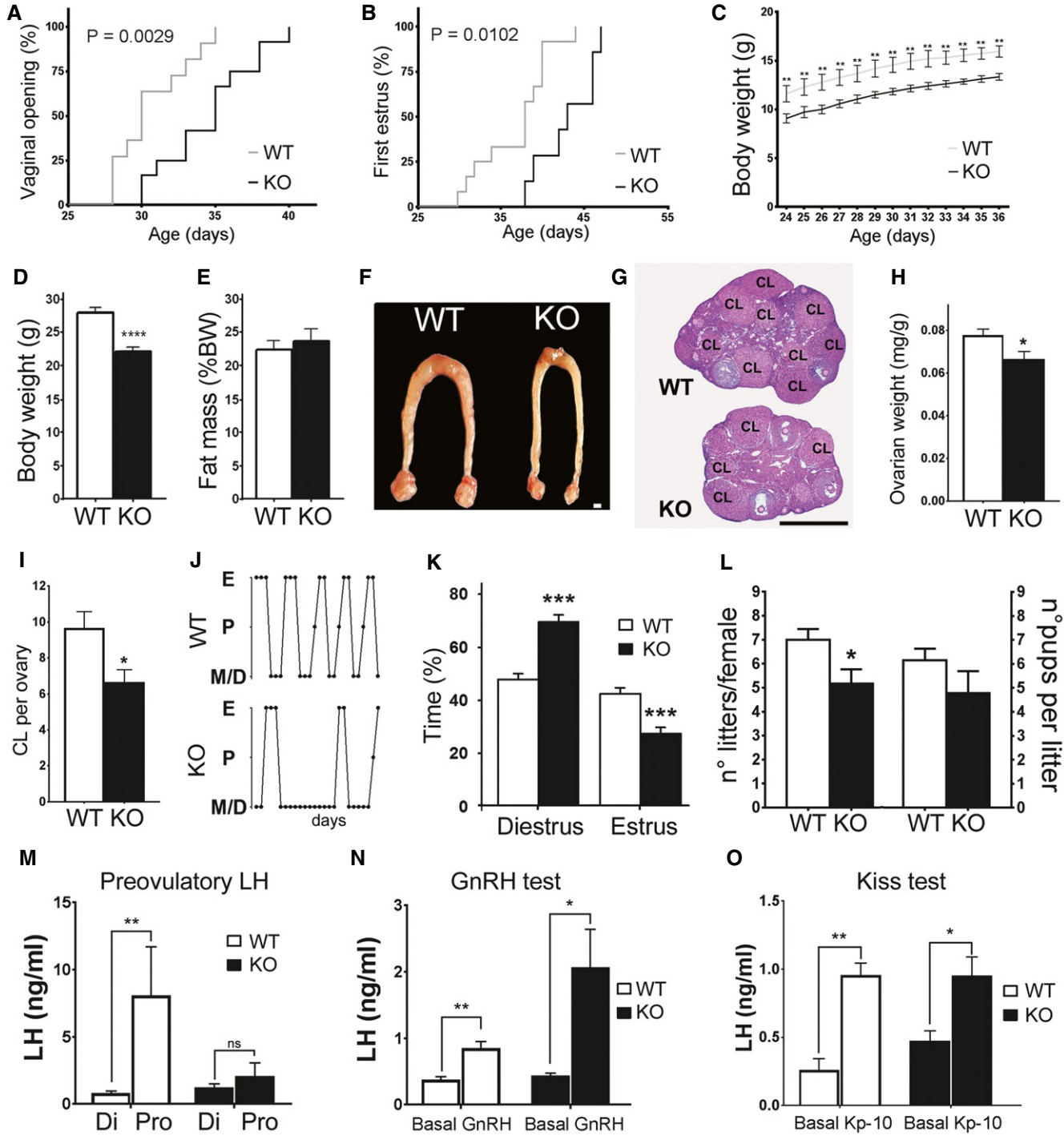

**Figure 4.**

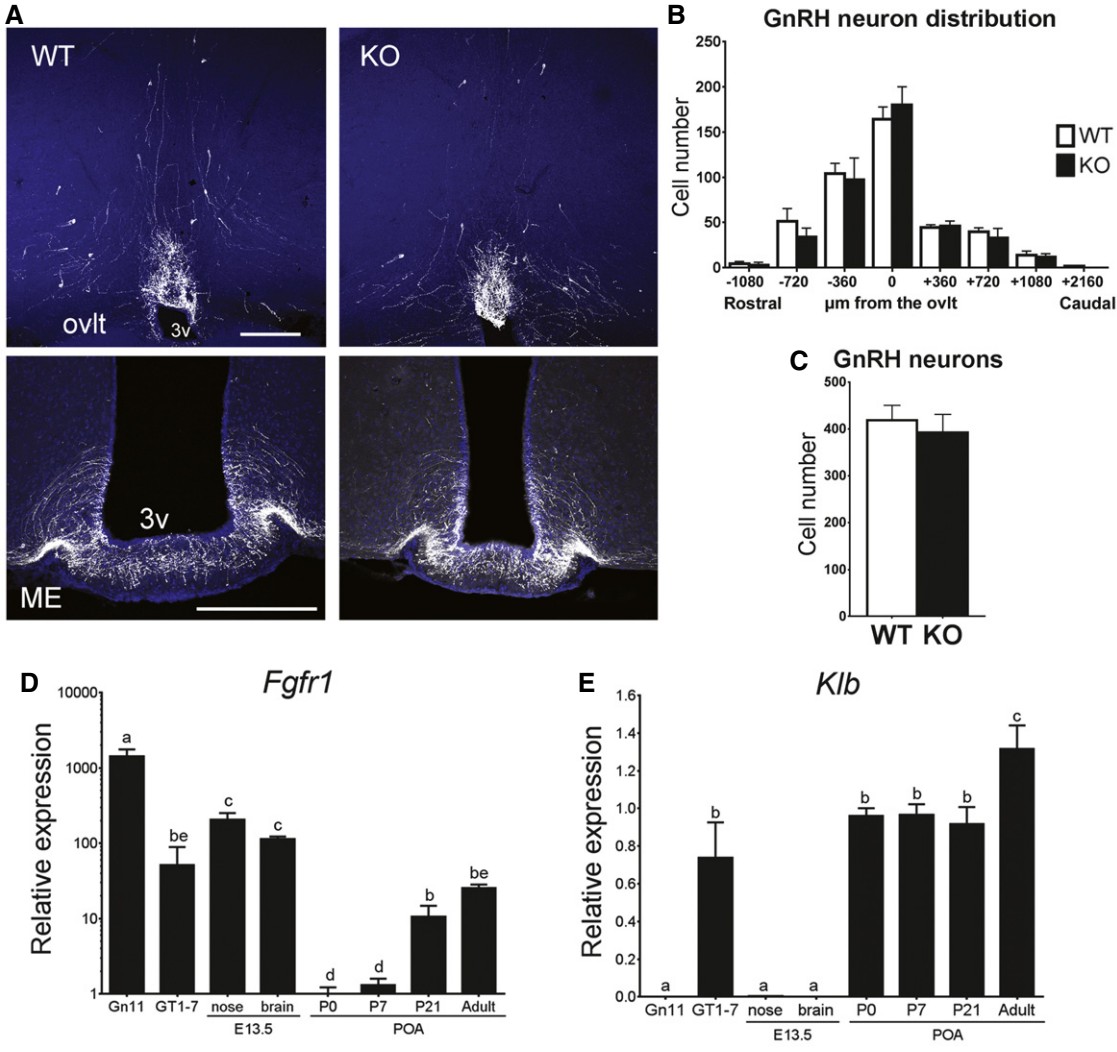

**Figure 5.  Intact positioning of GnRH neurons in KlbKO mice.**

A     Representative brain sections from wild-type and KlbKO adult females showing GnRH neuron cell bodies and fibers at the level of the ovlt in the POA (upper panel) and at the level of the median eminence in the mediobasal hypothalamus (ME, lower panel) level. Scale bars, 200 μm.

B     Analysis of GnRH neuron distribution along the rostro-caudal axis from the level of the medial septum/diagonal band of broca (MS/DBB, −1,080 μm) through the organum vasculosum of the lamina terminalis (ovlt, 0 μm), to the level of the median eminence (ME, +2,160 μm). WT: *n = 7*; KO: *n = 4*.

C     Quantification of total hypothalamic GnRH neurons in adult KlbKO females and WT littermates (WT: *n = 8*; KO: *n = 4*). Differences in GnRH cell number between groups were assessed using unpaired *t*-test.

D, E  Gene expression profiles in immature (Gn11) and mature (GT1-7) immortalized GnRH neurons and dissected tissues at embryonic day E13.5 (nose and brain) and different postnatal ages from postnatal day 0 (P0) to adulthood (*n ≥ 3* per group). Differences between groups were assessed using one-way ANOVA followed by Fisher's LSD test.

Data information: Values shown are mean ± SEM; different letters (abcde) indicate significant differences between groups (*P < 0.05*).

in GnRH vesicular pool at the nerve terminals in KlbKO mice (Fig EV4C), indicating that GnRH peptide synthesis and transport are not affected. These results indicate that *Klb* deficiency cause a defect in GnRH neuron homeostasis rather than development.

We further characterized the spatiotemporal expression pattern of *Fgfr1* and *Klb* in embryonic and postnatal tissues critical for GnRH neuron biology, as well as in immortalized GnRH cell lines (Gn11 and GT1-7 cells; Fig 5D and E). As expected, *Fgfr1* expression was high in the immature Gn11 cells as well as in embryonic

tissues. Postnatally, *Fgfr1* expression increased in the POA at weaning and persisted into adulthood (Fig 5D). Notably, *Klb* expression did not parallel *Fgfr1* expression, in that it was very low to absent in the embryonic brain and nose (Fig 5E). In contrast, higher levels of *Klb* were identified in the POA at birth and persisted until weaning with a further increase in adulthood (Fig 5E). These results further support a role of FGF21/KLB/FGFR1 signaling in the postnatal modulation of GnRH biology rather than embryonic development, which has previously been associated with FGF8/FGFR1 signaling.

## FGF21 stimulates neurite outgrowth in mature GnRH neurons *in vitro* and induces GnRH secretion in ME explants *ex vivo*

The cyclic release of GnRH peptide during the ovarian cycle in the pituitary portal system requires periodic neuroglial remodeling by molecules such as semaphorins and nitric oxide in the adult ME (De Seranno *et al*, 2004; Prevot *et al*, 2010; Giacobini *et al*, 2014; Parkash *et al*, 2015). FGF21 has recently been shown to be able to promote neurite outgrowth *in vitro* (Huang *et al*, 2013).

Consistently, we found that recombinant FGF21 (rFGF21) increased neurite length in mature immortalized GnRH neurons (GT1-7) *in vitro* (Fig 6A–C). To investigate the possibility that FGF21 may also act directly on GnRH neurons to modulate their function *in vivo*, we isolated GnRH neurons from adult *Gnrh::gfp* mice using fluorescence-activated cell sorting (FACS) and analyzed *Klb* expression. A twofold enrichment of *Klb* transcripts was found in GnRH neurons (Fig 6D). To further explore the role of FGF21 on GnRH neuronal function, we studied the effects of rFGF21 treatment on

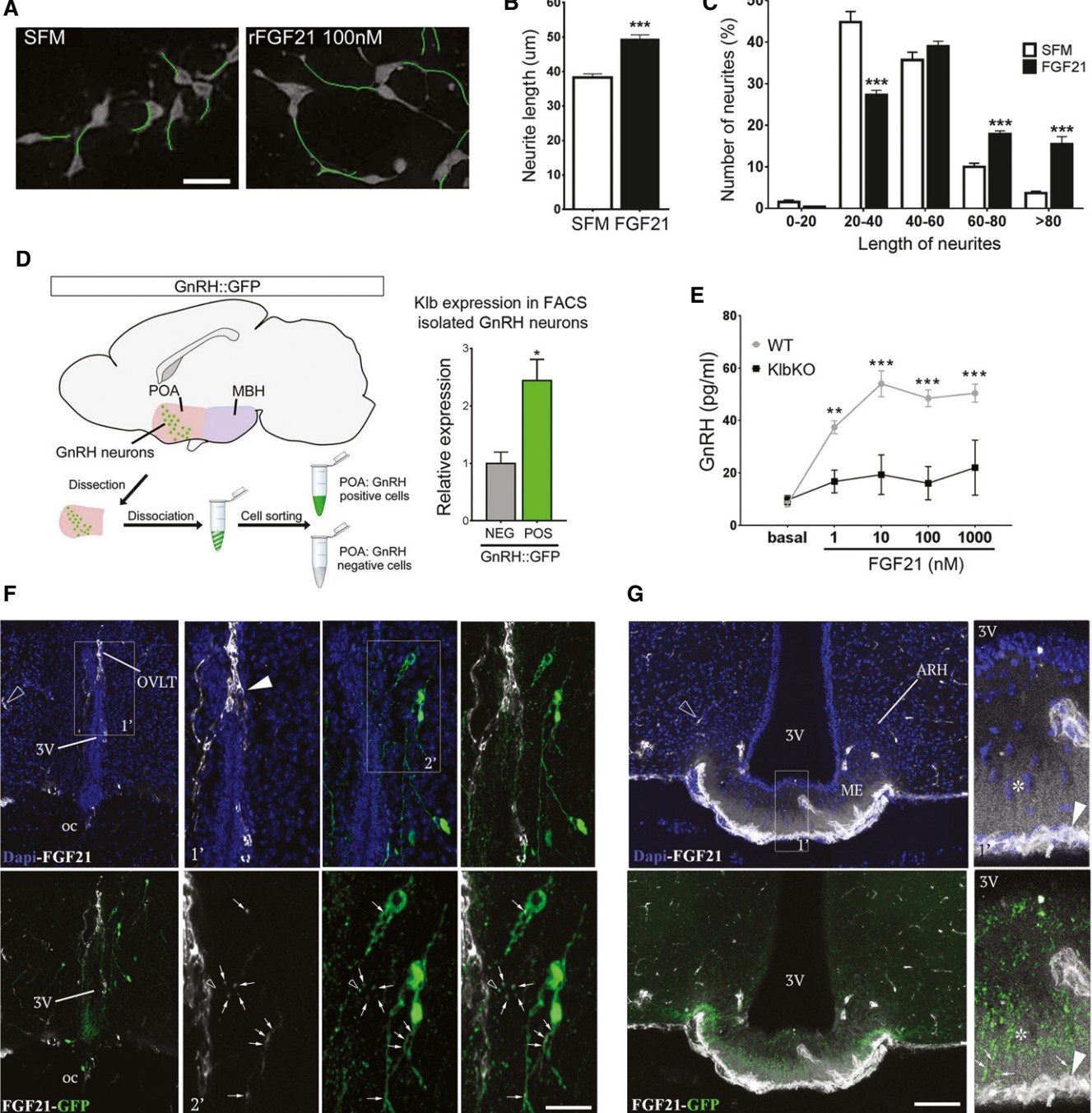

**Figure 6.**

GnRH secretion in living ME explants obtained from WT and KlbKO mice. Exogenous rFGF21 promoted robust GnRH release in a dose-dependent manner in WT mice, but not in KlbKO mice (Fig 6E). These results thus raise the possibility that peripheral FGF21 modulates GnRH secretion by acting directly on GnRH neuroendocrine terminals in the ME.

### Peripheral FGF21 reaches GnRH neurons through fenestrated vessels of ME or OVLT *in vivo*

We next examined whether peripheral FGF21 could reach hypothalamic GnRH neurons *in vivo*. Fluorescently labeled rFGF21 was injected intravenously to *Gnrh::gfp* mice 1 min before sacrifice. At this short interval, fluorescent rFGF21 was seen in the parenchyma of the ME and the organum vasculosum of the lamina terminalis (OVLT; Figs 6F and G, and EV5), where GnRH nerve terminals and GnRH dendrites are, respectively, distributed and extend beyond the blood–brain barrier (BBB; Ciofi *et al*, 2009; Herde *et al*, 2011). While circulating fluorescent rFGF21 remained trapped in the capillaries of the BBB (empty arrowheads, Fig 6F and G), it extravasated from the fenestrated vessels of the OVLT and the ME (Figs 6F and G, and EV5). GnRH neuroendocrine terminals in the external zone of the ME were seen to be surrounded by bloodborne fluorescent rFGF21 flooding the ME (arrows, Figs 6G and EV5). GnRH dendrites in the OVLT were found to be able to capture this circulating fluorescent rFGF21 and transport it toward the respective GnRH cell bodies (arrows, Figs 6F and EV5). Altogether, morphological data suggest that hypothalamic GnRH neurons can readily have access to peripheral FGF21.

## Discussion

Reproductive fitness is known to be tightly linked to energy availability, although the exact molecular mechanisms are still unknown (Bronson, 1986; Cahill, 2006). The hepatokine FGF21, a key central metabolic regulator (Liang *et al*, 2014; Owen *et al*, 2015), may constitute such a link between metabolism and reproduction. Our

finding of *KLB* mutations impairing FGF21 signaling in patients with congenital GnRH deficiency with a high frequency of associated metabolic defects supports this concept. Studies of *Klb*-deficient mice, which exhibit delayed puberty and subfertility partly due to a hypothalamic defect, are consistent with a role of Fgf21 in regulating reproduction. A previous report showed that *Fgf21* transgenic mice exhibit GnRH deficiency with infertility by repressing the vasopressin–kisspeptin pathway (Owen *et al*, 2013). Thus, both deficiency and excesses of FGF21 may lead to defects in GnRH function, consistent with the concept that a tightly regulated energy balance is required for optimal reproductive capacity.

While no CHH patients were found to harbor mutation in *FGF21*, 4% exhibit heterozygous mutations in *KLB*. All KLB mutants were confirmed to be loss of function *in vitro* as well as *in vivo* using a rescue assay in *C. elegans*. Notably, the p.F777delF mutation is found in seven probands of European descent. Although this variant is present in one out of 133 non-Finnish European controls (0.7%) in the ExAC database, it is significantly enriched in our CHH cohort versus controls. As most mutations in CHH are private, this occurrence suggests either a mutational hot spot or a founder effect (Avbelj Stefanija *et al*, 2012; Choi *et al*, 2015). The clinical spectrum ranges from severe GnRH deficiency with micropenis and cryptorchidism to milder forms such as CHH with reversal or fertile eunuch syndrome. This phenotypic variability is also observed among the seven CHH probands harboring the p.F777delF mutation, suggesting that other genetic or environmental factors may contribute to the phenotype. Additional mutations in other CHH genes were indeed found in five out of 13 patients (38%) including three patients with the p.F777delF mutation; this is compatible with an oligogenic model of inheritance that has been firmly established for other CHH genes (Sykiotis *et al*, 2010; Quaynor *et al*, 2011; Miraoui *et al*, 2013). To date, there remain 50% of CHH patients for whom no pathogenic mutation is known (Boehm *et al*, 2015), suggesting that additional mutations in currently unknown genes remain to be discovered.

Metabolic defects have been previously described in CHH patients and are thought to be secondary to sex steroid deficiency leading to unfavorable changes in body composition and inflammation (Zitzmann, 2009). Indeed, these metabolic parameters usually

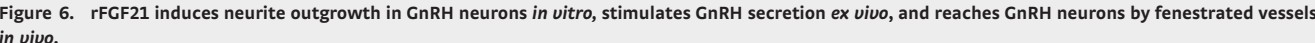

**Figure 6.  rFGF21 induces neurite outgrowth in GnRH neurons *in vitro*, stimulates GnRH secretion *ex vivo*, and reaches GnRH neurons by fenestrated vessels *in vivo*.**

A  Representative images of GT1-7 immortalized GnRH neurite traces after 48 h of culture in control medium (SFM) alone or with 100 nM rFGF21 (*n* = 4 per group). Scale bar 30 μm.

B  Average of the longest neurite length of GT1-7 cells in aforementioned conditions, unpaired *t*-test (*n* = 4 per group).

C  Neurite length distribution histogram showing significant increase of cells with longer neurites (> 80, 60–80 μm) and reduced number with shorter neurites (20–40 μm) after rFGF21 treatment, unpaired *t*-test (*n* = 4 per group).

D  Isolation protocol of GnRH neurons from GnRH::GFP mice by fluorescence-activated cell sorting (FACS) and *Klb* mRNA expression quantification in GnRH-positive (POS) and GnRH-negative (NEG) hypothalamic FACS-isolated cells (*n* = 3 per group), unpaired *t*-test. POA, preoptic area of the hypothalamus.

E  GnRH peptide content in conditioned medium from ME explant cultures treated with increasing doses of rFGF21 (*n* = 5 per group) for 20 min. Data were analyzed by two-way ANOVA followed by Sidak's multiple comparisons test.

F  Representative photomicrographs of the preoptic region showing GnRH neurons (GFP, green) and blood vessels from the BBB (empty arrowhead) and the OVLT (arrowhead) labeled by fluorescent rFGF21 (5 nmol/animal, white staining). Note the presence of white staining in GFP-expressing neuronal processes, dendrites, and cell body (arrows, inset 1′ and 2′). Nuclear contrast staining: DAPI; 3V, third ventricle; OVLT, organum vasculosum of the lamina terminalis; oc, optic chiasma. Scale bar: 150 μm for inset 1′, 40 μm for inset 2′, and 300 μm for other panels.

G  Representative photomicrographs showing blood vessels from BBB (empty arrowhead) and in ME (arrowhead) labeled by fluorescent rFGF21 (5 nmol/animal, white staining) in the tuberal region of the hypothalamus. Note that fluorescent rFGF21 administered intravenously diffuses into ME tissue (white staining, asterisk) where GnRH neuroendocrine axons are distributed (green, arrows). Nuclear contrast staining: DAPI; ME, median eminence; ARH, arcuate nucleus of the hypothalamus. Scale bar: 50 μm for inset 1′ and 130 μm for other panels.

Data information: Values shown are mean ± SEM; *$P$ < 0.05, **$P$ < 0.01, ***$P$ < 0.001.

 

improve after initiation of sex hormone replacement therapy (Tripathy *et al*, 1998; Naharci *et al*, 2007; Bayram *et al*, 2016). Further, short-term discontinuation of androgen therapy in male CHH patients leads to mild insulin resistance and increased glucose levels (Yialamas *et al*, 2007). Thus, unlike neurosensory hearing loss or anosmia, metabolic defects are not typically considered as primary events in CHH, apart from syndromic CHH caused by *LEP*, *LEPR*, or *PCSK1* mutations resulting in both GnRH deficiency and morbid obesity (Jackson *et al*, 1997; Farooqi *et al*, 2002, 2007). Leptin deficiency or resistance leads to a lack of metabolic cues that signal to the hypothalamus to regulate both energy homeostasis and fertility (Chehab, 2014). Here, we show that a majority of patients with *KLB* mutations (9/13) exhibit hypogonadotropic hypogonadism with some degree of metabolic defect (i.e. overweight, insulin resistance, and/or dyslipidemia), consistent with the metabolic role of FGF21/KLB/FGFR1 pathway. However, systematic metabolic phenotyping at diagnosis and after sex steroid replacement is required to firmly establish this association.

While *Fgfr1* is widely expressed in the brain, *Klb* expression is more restricted spatially and temporally, thus delimiting Fgf21 signaling (Bookout *et al*, 2013). We found that *Klb* mRNA was barely detectable during embryonic development in the head and nose, but it increased significantly after birth and maintained in adult/mature GnRH neurons, supporting a postnatal role of FGF21/KLB/FGFR1 signaling in GnRH biology. Consistent with the temporal pattern of *Klb* expression, we found that KlbKO mice had normal embryonic GnRH neuron development but exhibited delay puberty, disrupted cyclicity, and subfertility with an impaired GnRH function. The reproductive defects in the KlbKO mice are less severe than the phenotypes observed in our CHH patients with heterozygous *KLB* mutations. This phenotypic discrepancy between human and mice has been previously reported for other CHH genes like *TACR3* and *NSMF* (Yang *et al*, 2012; Quaynor *et al*, 2015). Notably, a similar reproductive phenotype was observed in KlbHET mice, indicating that *Klb* haploinsufficiency causes reproductive defects. This finding provides further evidence for a pathogenic role of the heterozygous *KLB* mutations in human, likely through the mechanism of haploinsufficiency.

In the adult hypothalamus, cyclic GnRH secretion is modulated by an interactive plasticity involving GnRH neuron terminals, tanycytes, and endothelial cells through locally secreted molecules such as semaphorins and nitric oxide (De Seranno *et al*, 2004; Giacobini *et al*, 2014; Parkash *et al*, 2015). While the role of FGF21 in the GnRH system was previously reported to suppress GnRH upstream signals such as vasopressin and kisspeptin (Owen *et al*, 2013), recent studies have shown that FGF21 may play a role in neuronal protection and neuronal plasticity (Huang *et al*, 2013; Leng *et al*, 2015). Our findings that FGF21 promotes neurite outgrowth in mature immortalized GnRH neurons (GT1-7) and stimulates GnRH secretion in living ME explants suggest a novel role for FGF21 in controlling fertility by modulating GnRH neuron structural plasticity. Further studies are warranted to define the exact mechanism of FGF21 action on neuroglial–endothelial plasticity *in vivo* and its physiological relevance for the GnRH system. Induced pluripotent stem cell (iPSC)-derived GnRH neurons (Lund *et al*, 2016; Poliandri *et al*, 2017) from CHH patients with *KLB* mutations would be an excellent model to elucidate how the human KLB mutants affect GnRH secretion.

Similar to leptin-sensitive neurons in the arcuate nucleus (Djogo *et al*, 2016; Pan & Myers, 2016), GnRH neurons extend their dendrites and terminals outside the blood–brain barrier (Ciofi *et al*, 2009; Herde *et al*, 2011). Therefore, it is tempting to speculate that GnRH neurons can also directly sense circulating metabolic signals. Leptin and insulin apparently do not act in this way, because GnRH neurons do not express leptin receptors (Quennell *et al*, 2009) and GnRH neuron-specific deletion of the insulin receptor does not lead to any reproductive defects (Evans *et al*, 2014). Our diffusion studies show that peripheral Fgf21 does not freely cross the BBB, but rather extravasates quickly from the fenestrated capillaries in the ME and the OVLT to gain direct access to GnRH neuronal dendrites and terminals lying outside the BBB (Ciofi *et al*, 2009; Herde *et al*, 2011). Taken together with our findings that *Klb* and *Fgfr1* are expressed in GnRH neurons, and that FGF21 stimulates GnRH secretion in ME *ex vivo*, it is likely that GnRH neuron terminals outside the BBB may perceive peripheral FGF21 promptly and efficiently to adapt GnRH secretion according to the metabolic state of the individual. It will thus be interesting to evaluate the FGF21 gene network for mutations in patients with functional hypogonadotropic hypogonadism (i.e., hypothalamic amenorrhea (Gordon, 2010) and obesity-related hypogonadotropic hypogonadism; Giagulli *et al*, 1994), as environmental cues related to energy balance are critical for reproductive fitness.

# Materials and Methods

## Subjects

The cohort includes 334 unselected probands with CHH (227 KS and 107 normosmic CHH, with 3:1 male–female ratio) along with affected and unaffected family members when available. The diagnosis of CHH includes: (i) absent or incomplete puberty by age 16; (ii) low/normal gonadotropin levels in the setting of low serum testosterone/estradiol levels; and (iii) otherwise normal anterior pituitary function and normal imaging of the hypothalamic–pituitary area (Pitteloud *et al*, 2002). Olfaction was assessed by self-report and/or formal testing (Lewkowitz-Shpuntoff *et al*, 2012). Metabolic assessments include BMI, fasting glucose and insulin levels, homeostasis model of assessment of insulin resistance (HOMA-IR; Tam *et al*, 2012), and lipid profile at presentation. Additionally, one patient underwent a hyperinsulinemic euglycemic clamp as previously described (Pitteloud *et al*, 2005). The controls include 191 in-house reproductively healthy subjects of non-Finnish European origin (Raivio *et al*, 2009).

## Genetic studies

Genomic DNA was extracted from peripheral blood samples using previously described methods (Miraoui *et al*, 2013). The coding exons and intronic splice regions (≤ 6 bp from the exons) of *FGF21* (NM_019113.2) and *KLB* (NM_175737) were analyzed by exome (*n* = 83) or Sanger sequencing (*n* = 251). Exome capture was performed using the SureSelect All Exon capture (Agilent Technologies, Santa Clara, CA USA) and sequenced on the HiSeq2500 (Illumina, San Diego CA USA). Raw sequences (fastq files) were analyzed using an in-house pipeline that utilizes published algorithms: the Burrows-Wheeler Alignment tool (BWA; Li & Durbin, 2009) for mapping the reads to the human reference sequence

(GRCh37), and the Genome Analysis Toolkit (GATK; DePristo *et al*, 2011) for the detection of single nucleotide variants (SNVs) and insertion/deletions (Indels). The resulting variants for both exome and Sanger sequences were annotated using SnpEff version 4.0 (Cingolani *et al*, 2012) and dbNSFP version 2.9 (Liu *et al*, 2013) to calculate minor allele frequency (MAF) and protein functional predictions.

A putative pathogenic variant is defined as (i) non-synonymous rare variant with MAF < 1% in both our in-house controls (*n* = 191; Raivio *et al*, 2009) and ethnically matched controls from the Exome Aggregation Consortium (ExAC, http://exac.broadinstitute.org); (ii) nonsense, frameshift, inframe insertion/deletion, splicing variants (predicted to affect splicing by at least two out three programs including Human Splice Finder (Desmet *et al*, 2009), MaxEnt (Desmet *et al*, 2009), and NNSplice (Reese *et al*, 1997), or missense variants predicted to be damaging by either SIFT (Ng & Henikoff, 2001) and Polyphen-2 (Adzhubei *et al*, 2010). A putative pathogenic variant is confirmed as a mutation when shown to be loss of function in functional assays (see below). ACMG guidelines for interpretation of sequencing variants (Richards *et al*, 2015; Li & Wang, 2017) were also applied to assess the pathogenicity of identified variants.

All subjects were also screened for known CHH genes via Sanger (*n* = 251) or by exome sequencing (*n* = 83): [*KAL1* (NM_000216.2), *FGFR1* (NM_023110.2), *FGF8* (NM_033163.3), *PROKR2* (NM_144773.2), *PROK2* (NM_001126128.13), *GNRHR* (NM_000406.2), *GNRH1* (NM_000825.3), *KISS1R* (NM_032551.4), *KISS1* (NM_002256.3), *TACR3* (NM_001059.2), *TAC3* (NM_013251.3), *NSMF* (NM_001130969.1), *HS6ST1* (NM_004807.2), *DUSP6* (NM_001946.2), *FGF17* (NM_003867.2), *FLRT3* (NM_198391.2), *IL17RD* (NM_017563.3), and *SPRY4* (NM_030964.3)]. Variants were confirmed by Sanger sequencing of both strands and with duplicate PCRs, and are described according to HGVS nomenclature (den Dunnen & Antonarakis, 2000).

The clinical and genetic studies were approved by the Institutional Review Board of Partners Healthcare and the ethics committee of the University of Lausanne, and were conducted in accordance with the guidelines of the Declaration of Helsinki. All participants provided written informed consent prior to study participation. Our datasets were obtained from subjects who have consented to the use of their individual clinical and genetic data for biomedical research, but not for unlimited public data release. Therefore, we submitted the data to the European Genome-phenome Archive (EGAS00001002568), and researchers can apply for access to the raw data.

## Functional studies

### Generation of KLB expression constructs

Human *KLB* cDNA in pCR4-TOPO vector (Plasmid ID HsCD00341606, Harvard Medical School) was used to generate constructs with N-terminus flag-tagged and C-terminus HA-tagged KLB cDNA in pcDNA3.1 + vector, which were further used as template for mutagenesis using QuikChange II XL Site-Directed Mutagenesis kit (Agilent Technology).

### FGF8/FGF21 signaling studies

Signaling activity of FGFR1 and KLB mutants was assessed using the osteocalcin FGF response element luciferase in a L6 myoblast cell model. L6 cells (sourced from ATCC) were transiently transfected with N-terminus myc-tagged *FGFR1* cDNA as previously described (Raivio *et al*, 2009), and HA-tagged *KLB* cDNA expression vector was added to the DNA mix when appropriate. 24 h post-transfection, cells were treated with recombinant FGF8 or FGF21 (16 h, 0–20 nM, except for double mutant assay 0–10 nM) as indicated and subsequently analyzed for luciferase activity. In each experiment, FGFR1 and KLB mutant values were expressed as a percentage of the maximal WT response and plotted with three-parameter agonist dose–response curves using Prism software (version 7; GraphPad). Transfection assays were performed in triplicate and repeated three times. The activity of each KLB/FGFR1 mutant was compared to WT in terms of $EC_{50}$ dose and the maximal FGF response.

### Co-immunoprecipitation assay

HEK293T cells (sourced from ATCC) were transiently transfected with N-terminus flag-tagged KLB and/or myc-tagged FGFR1 using FuGENE® 6 Transfection Reagent (Promega, Madison, WI, USA) according to the manufacturer's protocol. 48 h after transfection cells were lysed in lysis buffer (50 mM Tris–HCl pH8, 150 mM NaCl, 2 mM EDTA, 1% Triton X-100) supplemented with a cocktail of protease inhibitors (Thermo Fisher Scientific, Waltham, MA, USA). Clarified lysates were immunoprecipitated using monoclonal anti-flag (Sigma-Aldrich, Saint Louis, MO, USA) according to the manufacturer's instructions. Precipitated proteins were processed as previously described (Bouilly *et al*, 2014). KLB was revealed using mice monoclonal anti-flag (Sigma-Aldrich, Saint Louis, MO, USA), and FGFR1 was detected using mice monoclonal anti-myc (Millipore Corporation, Billerica, MA, USA). Experiments were repeated twice with consistent results.

### Ectopic protein expression and glycosylation analysis

The expression and maturation studies were performed in COS7 cells (sourced from ATCC) transiently transfected with C-terminus HA-tagged WT or mutated KLB constructs. As β-Klotho is an N-linked glycosylated protein, cell lysates were further subjected to enzymatic deglycosylation with peptide N-glycosidase F (PNGase F) and endoglycosidase H (EndoH), as described (Raivio *et al*, 2009). After deglycosylation, cell lysates were subjected to Western blotting analysis using anti-HA primary antibody (1:2,000; Sigma-Aldrich, Saint Louis, MO, USA) and rabbit anti-mouse HRP-conjugated secondary antibody (1:20,000, Invitrogen). Total protein abundance was calculated from the PNGase F-treated samples (~120 kDa), and the maturity index was calculated as the fraction of the mature (i.e. EndoH-resistant band, 140-kDa) to the total protein using the ImageJ program (Wayne Rasband, NIH, USA). Three independent experiments were performed in quadruplicate.

### Cell surface antibody binding assay

COS7 cells were transiently transfected with N-terminus flag-tagged KLB constructs and analyzed for KLB expression on the cell surface using an anti-flag antibody (1:1,500) and [$^{125}$I]-rabbit anti-mouse IgG (300,000 cpm/well; Perkin Elmer, Waltham, MA, USA) as previously reported (Raivio *et al*, 2009). Samples were run in quadruplicate and the experiment repeated three times.

### *Caenorhabditis elegans* studies

*Caenorhabditis elegans* strains were maintained at 20°C as described (Brenner, 1974). The single mutant alleles *klo-2* (*ok1862*; Polanska *et al*, 2011) and *klo-1* (*ok2925*) were generated by the *C. elegans* Knockout Consortium, and strains carrying these alleles were obtained from the Caenorhabditis Genetics Centre (CGC), and backcrossed at least three to five times prior to analysis. Both alleles are presumed null. TC446 is a double mutant of *ok1862* and *ok2925* and was generated by standard genetic crossing.

The huKLB constructs were derived from the plasmids pTK37 (p*klo-1*::gfp) and pTK53 (p*klo-2*::gfp; Polanska *et al*, 2011). Sequences encoding for wild-type human KLB and the F777del variant cDNAs were cloned downstream of *klo-1* and *klo-2* promoter sequences, removing sequences encoding for GFP present in the original plasmids pTK37 and pTK53, but retaining the *unc-54* 3'UTR. Since both promoters gave similar results, the rest of the mutant human KLB cDNAs were cloned similarly downstream of the *klo-1* promoter in pTK37. All DNA constructs were confirmed by sequencing. Transgenic arrays were generated using standard germ line transformation techniques (Mello *et al*, 1991). DNA constructs were injected to *klo-2* (*ok1862*); *klo-1* (*ok2925*) double mutant *C. elegans*, at 20 ng/μl with P*tph-1*::mCherry at 40 ng/μl as injection marker and pBluescript. Two to five independent transgenic lines were analyzed for each DNA construct. Fluorescent and DIC images were acquired using Zeiss AxioCam MRm camera mounted on Zeiss Axioskop2 or Zeiss Axioimager Z1 microscopes equipped with epifluorescence and DIC optics. Images were captured using Axiovision/Zen and further cropped and scaled using Adobe Photoshop CS4.

### Mouse studies

#### Housing

The generation and genotyping of KlbKO mice, kindly provided by Prof. Nabeshima, was previously described (Ito *et al*, 2005), and all animals used in this study were backcrossed onto a pure C57BL/6J background for more than 10 generations. KlbKO mice were obtained from heterozygous breeding pairs, and KlbHET mice were obtained from heterozygous wild-type breeding pairs. After weaning at 21 days of age, all mice were housed at room temperature (22°C) with a 12-h-light/12-h-dark cycle and free access to water and food.

#### Reproductive assessment

To assess puberty, post-weaning female mice were inspected daily to measure body weight and determine vaginal opening. Day of first estrus was determined after vaginal opening by daily cytological analysis of vaginal smears under an inverted microscope.

To study reproductive capacity, estrous cyclicity was monitored in 3-month-old females for 4 consecutive weeks by daily cytological analysis of vaginal smears. Blood samples were collected at different estrous cycle stages for LH measurement. To study fertility, a continuous mating protocol was used to measure the number of litters per females and litter size on KlbKO and WT littermates housed in a cage with a confirmed fertile WT male for 7 months. The reproductive capacity of KlbHET mice was assessed by a short-term fertility test. Briefly, WT and KlbHET females housed together (1:1) were exposed to a confirmed fertile male for 2 nights. To

induce proestrus with a preovulatory LH surge, KlbKO and WT females were exposed to bedding enriched with sexually experienced male urine, and sacrificed the evening of the third day of exposure. Blood was collected by intracardiac puncture and immediately placed on ice until plasma separation, and then stored at −80°C until LH measurement by ELISA.

Females underwent GnRH and Kisspeptin testings on the day of first diestrus, and blood samples were collected before and after IP injection of either 0.25 μg GnRH or 1 nmole Kisspeptin (Kp-10) per mouse. Briefly, 4 μl blood was collected by tail puncture, diluted in PBS Tween 0.05%, immediately frozen on dry ice, and stored at −80°C until LH measurements by ELISA.

To assess ovarian morphology, ovaries from 3- to 6-month-old KlbKO and WT littermates were collected and weighed, fixed by immersion in 4% paraformaldehyde (PFA) solution and stored at 4°C until tissue processing. Paraffin-embedded ovaries were serially sectioned at 5 μm thickness and stained with hematoxylin–eosin (histology facility, University of Lausanne, Vaud, Switzerland). Fresh and regressing corpora lutea were counted in every 5th section of an ovary by comparing the section with the preceding and following sections.

#### Metabolic assessment

Food intake was measured for mice placed in individual cages during at least 3 consecutive days. Lean mass, fat mass, and percent fat were determined on awake mice using an Echo MRI analyzer (Whole Body Composition Analyzer, Echo medical systems, Houston, TX, USA). Females underwent a glucose tolerance test (GTT) after an overnight fast. Glucose (2 mg/g b.w.) was administered intraperitoneally, and glucose levels were determined in tail blood using an ACCU-CHEK® Aviva blood glucose monitor and appropriate test strip (Roche Diagnostics). Oxygen consumption ($VO_2$), carbon dioxide production ($VCO_2$), and spontaneous locomotor activity were determined using indirect, open-circuit calorimetry in an Oxymax Metabolic Chamber system (Columbus Instruments, Columbus, OH, USA). The mice (with free access to food and water) were housed in separate chambers for 48 h for acclimatization before starting 48 h measurements. The average $VO_2$ was calculated during 48 h—two light cycles and two dark cycles (Somm *et al*, 2005, 2014).

#### Biochemical studies

Plasma total, HDL-, and LDL-cholesterol levels were analyzed using the Cobas C111 automated platform (Roche Diagnostics). Plasma insulin and leptin were measured with the metabolic Milliplex kit (MADKMAG-71K, Merck Millipore). LH was measured with an in-house, sensitive, two-site sandwich immunoassay as previously described (Steyn *et al*, 2013).

#### GnRH neuron neuroanatomical analysis

Brains were obtained from adult females by decapitation under deep general anesthesia (100 mg/kg of ketamine HCl and 10 mg/kg xylazine HCl) and preserved as previously described (Messina *et al*, 2011). Coronal hypothalamic sections were immunofluorescently stained for GnRH neurons using rabbit anti-GnRH (1:1,000; Immunostar) as primary antibody following a previously reported protocol (Messina *et al*, 2011). Confocal immunofluorescent imaging was performed using an inverted microscope (Zeiss LSM 780 Quasar), and images were processed

using ImageJ and Adobe Photoshop CS5. All slides were coded to blind sample identification prior to analysis of GnRH neurons number and distribution. Neurons were manually counted in each section along the rostro-caudal axis from the medial septum to the ME. For quantification of GnRH neuron distribution, serial sections were aligned at the OVLT as a common anatomical landmark between samples.

*Gene expression analysis*
Brain tissues were micro-dissected on a standard stereo-microscope under RNase-free conditions. Total RNA from tissues and cell cultures was extracted using TRIzol® reagent (Invitrogen). Standard protocols were followed for quantifying gene expression. Primers for detecting *Hprt, Rps29, Klb, Fgfr1, Gnrh1, Lepr, Kiss1r, Kiss1, Pomc, Avp, and Npy* are available in Appendix Table S2. GnRH neurons were isolated by fluorescence-activated cell sorting (FACS) from adult *GnRH::GFP* transgenic mice and gene expression performed as previously reported (Messina *et al*, 2016) using exon-boundary-specific TaqMan Gene Expression Assays (Applied Biosystems): *Gnrh1* (Gnrh1-Mm01315605_m1), *Klb* (Klb-Mm00473122_m1). Control housekeeping genes: *r18S* (18S-Hs99999901_s1); *Actb* (Actb-Mm00607939_s1).

*Median eminence explant cultures and GnRH secretion determination*
Adult female mice were sacrificed on the day of diestrus and ME explants were dissected and processed as previously described (Prevot *et al*, 1999). Conditioned medium was collected after explants were treated with increasing doses of rFGF21 (20 min) or with KCl 0.05 M in order to induce the release of the entire GnRH vesicle pool. Collected media were analyzed for GnRH content following a GnRH ELISA protocol (Phoenix Pharmaceuticals Inc., California, Catalog no. #FEK-040-02).

*Immortalized GnRH neuron culture and neurite outgrowth assay*
Gn11 and GT1-7 cells (Mellon *et al*, 1990) were cultured as previously described (Messina *et al*, 2011). For neurite outgrowth assay, GT1-7 cells were plated at low density ($2.5 \times 10^4$ cells/cm$^2$) on poly-L-lysine-coated 24-well culture plates and treated after adhesion with rFGF21. After 48 h, cells were fixed for 10 min in 4% PFA and stained with Coomassie brilliant blue prior to image acquisition. Neurite tracing was performed following recommended procedures using the ImageJ plugin, NeuronJ (Meijering *et al*, 2004; Meijering, 2010).

*Peripheral injections of fluorescent FGF-21*
Five nanomoles of fluorescent rFGF-21 (Cisbio Bioassays) were injected into the jugular vein of anesthetized *Gnrh::gfp* mice. Mice were decapitated 1 min after the injection, and the brain immersion fixed in 4% PFA (0.1 M PBS pH 7.4) for 2 h then placed in the same fixative + 20% sucrose overnight at 4°C. Brains were frozen in liquid nitrogen-cooled isopentane, and 30-micrometer-thick cryostat sections were collected on superfrost glass slides. Blood vessel staining was performed using rabbit anti-laminin antibody (Sigma AB19012, 1:100).

The mice experimental protocols were performed in accordance with the Swiss animal welfare laws under the authorization of the Service de la consummation et des affaires vétérinaire Vaud (no. VD 2659) and with the approval of Institutional Ethics Committees for the Care and Use of Experimental Animals of the Universities of Lille.

**The paper explained**

**Problem**
Reproduction is regulated by the hypothalamic secretion of gonadotropin-releasing hormone (GnRH). Absence of GnRH secretion or action results in congenital hypogonadotropic hypogonadism (CHH), a rare genetic disorder characterized by lack of puberty and infertility. Multiple causative genes have been identified for CHH including fibroblast growth factor receptor 1 (*FGFR1*). FGF21 signals through a receptor complex comprised of FGFR1 and β-Klotho (encoded by *KLB*) and acts as a potent regulator of metabolism. Given the close association between energy balance and reproductive fitness, we hypothesize that FGF21/KLB/FGFR1 signaling is implicated in GnRH biology and that mutations in FGF21/KLB/FGFR1 gene network underlie CHH.

**Results**
Combining human genetics, mouse models, and molecular biology, we identified that 13 of 334 (4%) of CHH probands harbor a heterozygous loss-of-function mutation in *KLB*. No mutations were identified in *FGF21*. The majority of CHH probands with *KLB* mutations exhibit metabolic defects. Mice deficient in *Klb* have delayed sexual maturation and impaired fertility due to a hypothalamic defect. FGF21 enhances GnRH release *in vitro*. Peripheral FGF21 have a potential to reach the hypothalamic GnRH neuron terminals residing outside of the blood–brain barrier.

**Impact**
We demonstrate that FGF21/KLB/FGFR1 signaling plays an essential role in GnRH biology and is a novel link between metabolism and reproduction in humans.

**Statistical analysis**

For *in vitro* and *in vivo* experiments, sample sizes were chosen according to the standard practice in the field. Unless otherwise indicated, all analyses were performed using Prism 7 (GraphPad Software), assessed for normality and variance, and log-transformed when appropriate. The analysis of differences between the allele frequency of *KLB* mutations in CHH patients and ExAC controls was performed by Fisher's exact test. For *in vitro* reporter gene assays, the activity of each FGFR1 or KLB mutant was compared to WT in terms of $EC_{50}$ dose and the maximal FGF response by *F*-test. Expression levels of WT and KLB mutants were compared using an unpaired *t*-test. In the *C. elegans* assay, a two-tailed Fisher's exact test was used to determine significance between human KLB WT vs. mutants rescue. For comparison of multiple groups with one or two independent variables, statistical significance was determined using one- or two-way ANOVA followed by Fisher's least significant difference *post hoc* analysis or Sidak's multiple comparisons test. For comparison between two groups of normally distributed data, a paired or unpaired *t*-test was used when appropriate. For comparison of event/time curves (i.e. timing of puberty), the Gehan–Breslow–Wilcoxon test was used. Pregnancy frequencies from the short-term fertility test were compared using chi-squared test. Unless otherwise indicated, the significance level was set at $P < 0.05$. Data groups are indicated as mean ± SEM. The exact *P*-values of each analysis are listed in Appendix Table S3.

**Expanded View** for this article is available online.

## Acknowledgments

We are grateful to the patients and families who contributed their time, medical information, and DNA samples to this study. We thank Prof. Nabeshima (Kyoto University School of Medicine, Kyoto, Japan) for providing the KlbKO mice, Sarah Gallet and Emilie Caron (Inserm U1172) for expert technical assistance, and Sue Stewart (Birmingham Women's and Children's NHS Foundation Trust) for the assistance in patient recruitment. This work was supported by the Swiss National Science Foundation Sinergia Grant (CRSII3_141960, N. Pitteloud, U. Albrecht and M. Mohammadi); the Swiss National Science Foundation grant (SNF 31003A 153328, N. Pitteloud); the Agence National pour la Recherche (ANR, France) Grant GlioShuttles4Metabolism (ANR-15-CE14-0025, V. Prévot); the grant BFU2014-57581-P (Ministerio de Economía y Competitividad, Spain; co-funded with EU funds from FEDER Program, M. Tena-Sempere); the National Institute of Dental and Craniofacial Research at the National Institutes of Health Grant (R01 DE-13686 to M. Mohammadi); the Harvard Reproductive Endocrine Sciences Center of Excellence in Translational Research in Reproduction & Infertility: The Eunice Shriver National Institute of Child Health and Human Development (NICHD P50 HD-28138, W. Crowley).

## Author contributions

NP, AM, VP, and MT-S designed research studies. CX, AAD, RQ, CDG, MD, VS, TRC, AAT, JMWK, and WFC participated in patient recruitment and the clinical studies. In addition, WFC contributed data and DNA samples from the Boston patient cohort. CX, JA, DC, GPS, and LP participated in the genetic study and data analysis. CX, HM, NJN, YS, and JB performed *in vitro* experiments and data analysis. TK performed *C. elegans* study and data analysis. AM, ES, UA, and VP performed mouse studies and data analysis. MM provided recombinant FGF21. NP, CX, AM, JA, GPS, AAD, MT-S, VP, and MM participated in writing the manuscript.

## Conflict of interest

The authors declare that they have no conflict of interest.

## For more information

**Web resources**

GnRH network, http://www.gnrhnetwork.eu/
ExAC database, http://exac.broadinstitute.org/
Ensembl, http://www.ensembl.org/index.html
UCSC Genome Bioinformatics, http://genome.ucsc.edu/
Genome Browser (NCBI), http://www.ncbi.nlm.nih.gov/sites/entrez?db=Genome
Mouse Genome Informatics website, http://www.informatics.jax.org/
Online Mendelian Inheritance in Man (OMIM), http://www.omim.org/
Uniprot, http://www.uniprot.org/
Polyphen-2, http://genetics.bwh.harvard.edu/pph2/
SIFT, http://sift.jcvi.org/
NNSplice, http://www.fruitfly.org/seq_tools/splice.html
Human Splice Finder, http://www.umd.be/HSF3/
InterVar, http://wintervar.wglab.org/

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
