## [Review Process File · EMBO Molecular Medicine]

KLB, encoding β -Klotho, is mutated in patients with congenital hypogonadotropic hypogonadism

Cheng Xu, Andrea Messina, Emmanuel Somm, Hichem Miraoui, Tarja Kinnunen, James Acierno Jr, Nicolas J. Niederländer, Justine Bouilly, Andrew A. Dwyer, Yisrael Sidis, Daniele Cassatella, Gerasimos P. Sykiotis, Richard Quinton, Christian De Geyter, Mirjam Dirlwanger, Valérie Schwitzgebel, Trevor R. Cole, Andrew A. Toogood, Jeremy M.W. Kirk, Lacey Plummer, Urs Albrecht, William F. Crowley Jr, Moosa Mohammadi, Manuel Tena-Sempere, Vincent Prevot & Nelly Pitteloud

Corresponding author: Nelly Pitteloud, Lausanne University Hospital

Review timeline:	Submission date:	01 December 2016
	Editorial Decision:	10 February 2017
	Revision received:	10 May 2017
	Editorial Decision:	13 June 2017
	Revision received:	23 June 2017
	Accepted:	27 June 2017

Editor: Céline Carret

Transaction Report:

1st Editorial Decision

10 February 2017

Thank you for the submission of your manuscript to EMBO Molecular Medicine and importantly, thank you for your patience while we completed the peer-review process. We have now heard back from the two referees whom we asked to evaluate your manuscript.

As you will see from the comments below, both reviewers appreciate the interest of the findings, however both have concerns about the pathogenic role of the heterozygous mutations described and a deeper analysis of the genotypes and putative bigenic role should be provided; referee 2 in addition, would like to see some structural modeling, better discussion and more details to help in defining some mechanism.

Given the balance of these evaluations, we feel that we can consider a revision of your manuscript if you can address the issues that have been raised within the space and time constraints outlined below. Please note that it is EMBO Molecular Medicine policy to allow only a single round of revision and that, as acceptance or rejection of the manuscript will depend on another round of review, your responses should be as complete as possible.

Please read below for important editorial formatting and consult our guidelines to accelerate the editorial process should your manuscript move forward.

I look forward to receiving your revised manuscript.

***** Reviewer's comments *****

Referee #1 (Remarks):

The authors previously identify an FGFR1 p.L342S mutation in a patient with Kallmann syndrome and identify FGF8 as a critical ligand for FGFR1 in GnRH biology during embryonic development. This patient also had metabolic phenotypes with severe insulin resistance. In post-natal biology FGF21 signals through FGFR1C in the presence of beta-Klotho. The authors suspect that defects in FGF21/KLB/FGFR1 signaling pathway may be involved in GnRH deficiency in humans and rodents. A candidate gene approach in 334 patients with congenital hypogonadotropic hypogonadism allowed the identification of 7 heterozygous mutations in KLB, six missense and one inframe deletion. Most patients exhibit metabolic defects. All deletions had a deleterious effect in vitro. Complementation studies in *C. elegans* deleted of the two homologs *klo-1* and *klo-2* showed that the mutants failed or had a decrease ability to rescue the cyst phenotype of the double deleted mutant. *Klb* deficient mice had delayed puberty and subfertility due to defects in GnRH secretion. KLB is expressed in the post-natal hypothalamus. FGF21 stimulates neurite outgrowth in mature GnRH neurons in vitro and induces GnRH secretion in median eminence (ME) explants ex vivo. Peripheral FGF21 reaches GnRH neurons through fenestrated vessels of ME or vascular organ of lamina terminalis in vivo.

This is an interesting work with many different molecular and cellular approaches in vitro and in vivo which demonstrate convincingly the role of KLB in GnRH biology.

However I have questions concerning the pathogenic role of the mutations described in this study:

1) Are these heterozygous mutations sufficient to impair KLB function in vitro and in vivo? Indeed one allele is not mutated in the patients studied. The dominant negative effect of the mutations has not been demonstrated. Increasing amounts of mutated versus wild type KLB expression vectors could highlight such an effect in vitro. Alternatively the authors have not shown the existence of haploinsufficiency. This is critical to establish a direct link between these heterozygous mutations and a pathogenic effect in IHH.

Animal models do not evidence a role for heterozygous mutants: Overexpression in *C. elegans* have not been performed in the context of a normal allele. Complementation in the context of deleted *Klb* *C. elegans* does not reproduce the heterozygous genotypes of the patients. Furthermore a partial rescue of the cyst phenotype is nevertheless observed with four mutants in *klo-2* and *klo1* deleted *C. elegans*, which is in keeping with a partial loss of function of the mutants in this model. The homozygous *Klb* knock mice exhibit only a mild reproductive phenotype. What is the phenotype of heterozygous mice?

If neither dominant negative effect nor haploinsufficiency are found, CGH array should be performed to eliminate a deletion in the second allele not evidenced by Sanger analysis. Why do 83 patients had exome studies and 251 Sanger studies of FGF21 and KLB genes? Which patients had exome studies? Were *Klb* deletions eliminated in those cases?

2) In some cases the mutations are associated with mutations of other genes responsible for IHH and the authors suspect a digenic effect. Why should the same p.F777delF mutation be deleterious when associated with a PROKR2 mutation in one case and alone in a patient with a similar phenotype? In the absence of evidence of dominant negative effect of these heterozygous mutations or of haploinsufficiency the authors should discuss the study of other genes potentially involved. How many such patients had exome studies? What was the result?

Referee #2 (Remarks):

The manuscript by Xu et al reports variants of klotho-beta, the FGF21 co-receptor, as a potentially

new molecular genetic cause of congenital hypogonadotropic hypogonadism. They also characterize the role of klotho beta for reproductive maturation in a knockout mouse, and test the potential roles of FGF21 in hypothalamic GnRH-neurons. The paper is to my opinion interesting and expands knowledge of the roles of FGF21 in signaling and the FGFR1/KLB/FGF21 axis as a cause of CHH.

The authors hypothesized that a specific CHH-associated FGFR1 variant, known to affect FGF8 binding site, also would inhibit binding of beta-klotho (KLB), a necessary coreceptor for FGF21 binding to FGFR1. FGF21 did not show mutations, but in sequencing of 334 patients with CHH, they identified 13 probands with seven different heterozygous rare variants in KLB. Seven probands carried a "hotspot" mutation of F777delF, being enriched to this patient population (frequency 0.2 in CHH vs 0.001 in ethnically matched population). They test these variants in a cell-based reporter system and in *c. elegans* model rescue assay, implicating variable functional defects of the putative mutations, supporting the contributing role of most of them in CHH. These data are quite intriguing.

Comments:

- 1) In their reporter system, the identified KLB variants showed decreased maximal response, with the exception of 309W variant, which is also poorly conserved; *Xenopus* carries the variant W at this site, suggesting it to be a neutral change. Despite some functional findings with this variant (*c. elegans* complementation defective), the lack of conservation and mild functional effects speak against this variant to pathogenic. Similarly, the 574T variant shows little conservation and little functional effects. However, other variants affect highly conserved sites, appear as functionally defective, and therefore as potentially pathogenic. Please revise the text to indicate the likely neutral vs likely pathogenic variants.
2. They find KLB variants to exist in some patients in combination with known CHH/Kallman sdr variants and state: "This finding indicates that some degree of oligogenicity is present in this population." - Just the co-occurrence does not indicate, but at best it may suggest a bigenic contribution. It still remains a possibility that the KLB variants are not required for the phenotype in the case other mutations exist. Are the patients with potential bigenic alleles more severely affected than those that carry single alleles?
3. One subject had KLB F777delF and FGFR1 R78C, which they coexpressed in their reporter assay, showing synergistic effect. In other assays they used the FGFR1 L342 mutant. Do the authors predict that L342 and R78 are close to each other in 3D structure, so that both of them would affect klotho binding? Structural modeling of the mutations to FGFR1 structure would be useful, to illustrate the potential binding site.
4. The second part of their study is the characterization of KLB-KO mice, which are found to have delayed reproductive maturation, nicely fitting to the role of KLB in CHH - but not indicating a link. They find no developmental defects in the organization of GnRH-neurons, or any changes in gonadotropin metabolism, and therefore they conclude that the potential effect of KLB/FGF21 for hypothalamus would be homeostatic. They then go on to study how FGF21 can reach hypothalamic GnRH neurons from the periphery and show that FGF21 stimulates GnRH release from hypothalamus, and that in KLB-KO this does not occur. Whether their identified KLB variants blunt the release, remains open. Any cell culture studies with patient mutations (iPSC derived CHH patient neurons) and their responses to FGF21 would shift the evidence to a completely different level. I do acknowledge that such materials are not easy to get, and therefore this is not a requirement for further experimentation.
5. Figure 5F-G present how fluorescent FGF21 can extravasate specifically in the area of median eminence where BBB is not tight and the GnRH-positive neurons become in contact with periphery. I really tried to see what the authors aim to show in 5F, but the findings were not at all evident from the figures. If they wish to show that FGF21 is first inside the vessels/capillaries and then extravasates, they should at least use a vascular fluorescent marker to point the sites of capillaries vs parenchyma. The labeling in the figure "fluorescent FGF21 vs fluorescent FGF21-GFP" is not clear. GFP is in GnRH neurons, right? Mark this clearly. What is the blue staining?
6. The result that FGF21 stimulates GnRH release would suggest that the liver - the main metabolic organ secreting FGF21 as a response to fasting (in mice, not as clearly in humans) - would regulate reproductive hormone signaling through hypothalamus. This is quite an intriguing possibility,

linking reproductive status to nutrient availability. Is there data suggesting that obese individuals show delayed sexual maturation (lack of fasting periods)? In humans the data suggests the opposite. A short speculation on the relevance of the findings to FGF21 expression and feeding vs. reproduction could be included.

Minor comment:

Terminology: instead of mutants, they should call KLB-changes as variants. This study reports potential variants in CHH, but especially the potential bigenic roles remain to be verified.

1st Revision - authors' response

10 May 2017

We thank you and the Reviewers for the diligent and insightful review of our manuscript. We have responded to each comment and have adapted the manuscript accordingly. The changes are marked in blue in the revised manuscript. We hope that you will find the revised version suitable for publication in *EMBO Molecular Medicine*.

Thank you very much for your kind consideration of our work.

Responses to Reviewers

Referee #1:

The authors previously identify an FGFR1 p.L342S mutation in a patient with Kallmann syndrome and identify FGF8 as a critical ligand for FGFR1 in GnRH biology during embryonic development. This patient also had metabolic phenotypes with severe insulin resistance. In post-natal biology FGF21 signals through FGFR1C in the presence of beta-Klotho. The authors suspect that defects in FGF21/KLB/FGFR1 signaling pathway may be involved in GnRH deficiency in humans and rodents. A candidate gene approach in 334 patients with congenital hypogonadotropic hypogonadism allowed the identification of 7 heterozygous mutations in KLB, six missense and one inframe deletion. Most patients exhibit metabolic defects. All deletions had a deleterious effect in vitro. Complementation studies in C. elegans deleted of the two homologs klo-1 and klo-2 showed that the mutants failed or had a decrease ability to rescue the cyst phenotype of the double deleted mutant. Klb deficient mice had delayed puberty and subfertility due to defects in GnRH secretion. KLB is expressed in the post-natal hypothalamus. FGF21 stimulates neurite outgrowth in mature GnRH neurons in vitro and induces GnRH secretion in median eminence (ME) explants ex vivo. Peripheral FGF21 reaches GnRH neurons through fenestrated vessels of ME or vascular organ of lamina terminalis in vivo.

This is an interesting work with many different molecular and cellular approaches in vitro and in vivo which demonstrate convincingly the role of KLB in GnRH biology.

We thank the Reviewers for the positive review of our study and have responded to each of the points below.

However I have questions concerning the pathogenic role of the mutations described in this study: 1) Are these heterozygous mutations sufficient to impair KLB function in vitro and in vivo? Indeed one allele is not mutated in the patients studied. The dominant negative effect of the mutations has not been demonstrated. Increasing amounts of mutated versus wild type KLB expression vectors could highlight such an effect in vitro. Alternatively the authors have not shown the existence of haploinsufficiency. This is critical to establish a direct link between these heterozygous mutations and a pathogenic effect in IHH. Animal models do not evidence a role for heterozygous mutants: Overexpression in C. elegans has not been performed in the context of a normal allele. Complementation in the context of deleted Klb C. elegans does not reproduce the heterozygous genotypes of the patients. Furthermore a partial rescue of the cyst phenotype is nevertheless observed with four mutants in klo-2 and klo1 deleted C. elegans, which is in keeping with a partial loss of function of the mutants in this model. The homozygous Klb knock mice exhibit only a mild reproductive phenotype. What is the phenotype of heterozygous mice? If neither dominant negative effect nor haploinsufficiency are found, CGH array should be performed to eliminate a deletion in the second allele not evidenced by Sanger analysis. Why do 83 patients had exome studies and 251

Sanger studies of FGF21 and KLB genes? Which patients had exome studies? Were Klb deletions eliminated in those cases?

This is an important comment. A dominant negative mode of action is indeed a feasible mechanism that could underlie the pathogenicity of the heterozygous *KLB* mutations. Through biochemical studies we have previously shown that KLB enhances FGF21-FGFR1c binding and hence promotes FGF21 signaling by simultaneously tethering FGF21 and FGFR1c to itself through two distinct sites (Goetz et al, 2012). If the loss-of-function mutations impaired KLB function without harming the FGF21 or FGFR1c binding sites then such mutant KLB alleles would competitively inhibit formation of KLB-FGFR1c-FGF21 ternary complex containing the wild type KLB allele, i.e. act in a classical dominant negative fashion. To test this possibility, we compared the binding of wild type and mutated KLB alleles with FGFR1c via immunoprecipitation experiments. All the KLB mutants coprecipitated with FGFR1c to the same extent as the wild type KLB, despite the fact they all incurred a loss in the ability to support FGF21 signaling as shown in the initial manuscript (Figure 1C-D). This finding is consistent with a dominant negative effect. We have added the new results in the revised manuscript (Page 8, Paragraph 3, Figure 1E).

As suggested by the Reviewer, we further investigated the reproductive phenotype of *Klb* heterozygous (*Klb*Het) female mice. We found that the *Klb*Het mice exhibit disrupted estrous cycles, blunted LH levels at estrus stage and impaired fertility (assessed by a short-term fertility test) – these phenotypes are similar to those of the *Klb*KO mice. These findings indicate that the phenotype of the *Klb*Het mice is due to haploinsufficiency. We have added the reproductive phenotype of the *Klb*Het mice as well as a discussion point in the revised manuscript (Page 13, Paragraph 2; Page 18, Paragraph 1; Figure EV4).

Thus, we conclude that in patients harboring heretozygous loss-of-function mutations in *KLB*, both mechanisms likely operate in parallel: a dominant-negative effect and a haploinsufficiency effect.

This explanation would be consistent with the partial rescue observed with some mutations in *C. elegans*, where, as the Reviewer correctly points out, no wild-type allele was present. Although the primary aim of the assay was to further test the functionality of *KLB* mutants (loss-of-function *in vitro*), we agree with the Reviewer that the results suggest mechanisms beyond an exclusive dominant-negative effect.

We performed Sanger sequencing in the 1st cohort (n = 251) and exome sequencing in the 2nd cohort (n = 83). Among the 13 *KLB* mutations, 11 were identified by Sanger and 2 by exome (this is now indicated in the revised Table 2). We further performed copy number variation calling in all available exomes; using the XHMM algorithm (Fromer et al, 2012), we did not identify any deletion within or around the *KLB* gene region. Furthermore, we evaluated *KLB* mRNA expression in lymphoblast cells from two available patients from the Sanger cohort (Subject 5 and Subject 7), both of whom harbor the *KLB* p.F777delF mutation. Sequencing the cDNA showed a heterozygous peak at the mutation site, demonstrating that both alleles are expressed at the mRNA level. Based on these findings, we conclude that a deletion of the second allele in *KLB* is unlikely to be a significant contributor in our cohort.

2) In some cases the mutations are associated with mutations of other genes responsible for IHH and the authors suspect a digenic effect. Why should the same p.F777delF mutation be deleterious when associated with a PROKR2 mutation in one case and alone in a patient with a similar phenotype? In the absence of evidence of dominant negative effect of these heterozygous mutations or of haploinsufficiency the authors should discuss the study of other genes potentially involved. How many such patients had exome studies? What was the result?

Indeed, all of the 13 CHH probands harboring *KLB* mutation were screened for other 18 major CHH genes either by Sanger sequencing or by exome sequencing. As outlined in the initial manuscript, 5 out of 13 patients have additional mutations in known CHH genes. Further, the fact that we have 7 patients with the same p.F777delF mutation allowed us to explore genotype-phenotype correlations. In agreement with the Reviewer's comment, we found phenotypic variation in these patients, which suggests that other factors, such as additional unknown genetic modifiers or environmental interactions, may contribute to the pathogenesis of CHH. The fact that genotype-correlations are not optimal is consistent with the generally appreciated fact that there remain 50% of CHH patients for

whom no pathogenic mutation is known (Boehm et al, 2015), suggesting that additional mutations in currently unknown genes remain to be discovered. These points have been added in the Discussion (Page 16, Paragraph 2).

Referee #2:

The manuscript by Xu et al reports variants of klotho-beta, the FGF21 co-receptor, as a potentially new molecular genetic cause of congenital hypogonadotropic hypogonadism. They also characterize the role of klotho beta for reproductive maturation in a knockout mouse, and test the potential roles of FGF21 in hypothalamic GnRH-neurons. The paper is to my opinion interesting and expands knowledge of the roles of FGF21 in signaling and the FGFR1/KLB/FGF21 axis as a cause of CHH.

The authors hypothesized that a specific CHH-associated FGFR1 variant, known to affect FGF8 binding site, also would inhibit binding of beta-klotho (KLB), a necessary coreceptor for FGF21 binding to FGFR1. FGF21 did not show mutations, but in sequencing of 334 patients with CHH, they identified 13 probands with seven different heterozygous rare variants in KLB. Seven probands carried a "hotspot" mutation of F777delF, being enriched to this patient population (frequency 0.2 in CHH vs 0.001 in ethnically matched population). They test these variants in a cell-based reporter system and in c. elegans model rescue assay, implicating variable functional defects of the putative mutations, supporting the contributing role of most of them in CHH. These data are quite intriguing.

We thank the Reviewer for the positive remarks and have responded to each the points below.

Comments:

1) In their reporter system, the identified KLB variants showed decreased maximal response, with the exception of 309W variant, which is also poorly conserved; Xenopus carries the variant W at this site, suggesting it to be a neutral change. Despite some functional findings with this variant (C. elegans complementation defective), the lack of conservation and mild functional effects speak against this variant to pathogenic. Similarly, the 574T variant shows little conservation and little functional effects. However, other variants affect highly conserved sites, appear as functionally defective, and therefore as potentially pathogenic. Please revise the text to indicate the likely neutral vs likely pathogenic variants.

We agree with the Reviewer that R309 residue is not conserved in Xenopus, but actually Xenopus carries N at this site rather than W, thus the difference is likely less dramatic compared to W (R being basic, N being polar and W being nonpolar). We apologize for the low resolution of Figure S1 and have provided figures with higher resolution in our revised manuscript. Further, we have shown that R309W exhibited a 3-fold increase in EC₅₀ compared to WT in the reporter assay (0.52 M vs 0.15 M, $p < 0.001$), consistent with loss-of-function.

We further assessed the KLB variants according to the guidelines of the American College of Medical Genetics and Genomics (ACMG) (Li & Wang, 2017; Richards et al, 2015), which integrate evidence from population data, computational algorithms (including conservation), functional assays and segregation data. All the KLB variants except p.L1011P were classified as pathogenic or likely pathogenic variants, while p.L1011P was classified as 'variant of uncertain significance'. We have added this information in the revised manuscript (Page 10, Paragraph 2; Table 1).

2. They find KLB variants to exist in some patients in combination with known CHH/Kallman sdr variants and state: "This finding indicates that some degree of oligogenicity is present in this population." - Just the co-occurrence does not indicate, but at best it may suggest a bigenic contribution. It still remains a possibility that the KLB variants are not required for the phenotype in the case other mutations exist. Are the patients with potential bigenic alleles more severely affected than those that carry single alleles?

We agree with the Reviewer that the constellation of two CHH gene defects is a suggestion of oligogenicity, but not a direct proof and we have corrected this sentence in the revised manuscript (Page 11, Paragraph 2).

Apart from the observation of additional mutations in 5 out of 13 (38%) CHH patients with KLB mutations, several other lines of evidence also suggest a digenic contribution in those cases: (i) the

other mutations in known CHH gene were all found in heterozygosity, and some are known to be insufficient to cause CHH alone – CHH associated with mutations in *GNRHR* and *PROKR2* is typically inherited in a recessive fashion (Avbelj Stefanija et al, 2012; de Roux et al, 1997); (ii) the cell-based reporter assay testing both KLB F777delF and FGFR1 R78C mutants (found in Subject 9) showed an additive effect on FGF21 signaling, providing *in vitro* evidence supportive of digenicity in this particular case.

Within the families with mutations in two different genes (Pedigree 4 and 5), we observed a more severe phenotype in the probands with digenic mutations compared to their parents with single gene defects. Among probands with *KLB* mutations, nine patients (9/13, 70%) exhibit severe GnRH deficiency. The remaining four patients have partial GnRH deficiency, 3 of whom carry heterozygous *KLB* mutation alone. These observations further suggest that the genetic load may impact the phenotypic severity. We have added this point in the revised manuscript (Page 11, Paragraph 2).

3. One subject had KLB F777delF and FGFR1 R78C, which they coexpressed in their reporter assay, showing synergistic effect. In other assays they used the FGFR1 L342 mutant. Do the authors predict that L342 and R78 are close to each other in 3D structure, so that both of them would affect klotho binding? Structural modeling of the mutations to FGFR1 structure would be useful, to illustrate the potential binding site.

We thank the Reviewer for this useful comment. FGFR1 L342 and R78 residues are not close to each other in the 3D structure. L342 is a key constituent of a hydrophobic groove in the 3rd immunoglobulin domain of FGFR1c that is critical for KLB binding (Goetz et al, 2012). By contrast, R78 lies in the 1st Immunoglobulin domain of FGFR1c which is dispensable for KLB interaction (Goetz et al, 2012). Consistently, the reporter assay studying the co-transfection of FGFR1 R78C and KLB F777delF showed an additive rather than a synergistic effect on FGF21 signaling. We have corrected this in the manuscript (Page 9, Paragraph 1).

4. The second part of their study is the characterization of KLB-KO mice, which are found to have delayed reproductive maturation, nicely fitting to the role of KLB in CHH - but not indicating a link. They find no developmental defects in the organization of GnRH-neurons, or any changes in gonadotropin metabolism, and therefore they conclude that the potential effect of KLB/FGF21 for hypothalamus would be homeostatic. They then go on to study how FGF21 can reach hypothalamic GnRH neurons from the periphery and show that FGF21 stimulates GnRH release from hypothalamus, and that in KLB-KO this does not occur. Whether their identified KLB variants blunt the release, remains open. Any cell culture studies with patient mutations (iPSC derived CHH patient neurons) and their responses to FGF21 would shift the evidence to a completely different level. I do acknowledge that such materials are not easy to get, and therefore this is not a requirement for further experimentation.

We thank the Reviewer for the insightful comment. We agree that our study did not specifically elucidate how the human KLB mutants affect GnRH secretion. Thus we agree with the Reviewer that iPSC-derived CHH patient neurons would be an excellent model to examine this, and we state so in the revised manuscript (Page 19, Paragraph 1).

5. Figure 5F-G present how fluorescent FGF21 can extravasate specifically in the area of median eminence where BBB is not tight and the GnRH-positive neurons become in contact with periphery. I really tried to see what the authors aim to show in 5F, but the findings were not at all evident from the figures. If they wish to show that FGF21 is first inside the vessels/capillaries and then extravasates, they should at least use a vascular fluorescent marker to point the sites of capillaries vs parenchyma. The labeling in the figure "fluorescent FGF21 vs fluorescent FGF21-GFP" is not clear. GFP is in GnRH neurons, right? Mark this clearly. What is the blue staining?

We thank the Reviewer for this comment. To better demonstrate the dynamic diffusion of FGF21 from the fenestrated vessels, we provide a new figure with the straining of laminin as a marker of the blood vessels (Figure EV5) and we clarified the labeling of Figure 5F-G and indicated that the blue staining is the nuclear stain DAPI.

6. *The result that FGF21 stimulates GnRH release would suggest that the liver - the main metabolic organ secreting FGF21 as a response to fasting (in mice, not as clearly in humans) - would regulate reproductive hormone signaling through hypothalamus. This is quite an intriguing possibility, linking reproductive status to nutrient availability. Is there data suggesting that obese individuals show delayed sexual maturation (lack of fasting periods)? In humans the data suggests the opposite. A short speculation on the relevance of the findings to FGF21 expression and feeding vs reproduction could be included.*

We thank the Reviewer for this comment. Large epidemiologic studies on the timing of puberty have shown an earlier secular trend of pubertal onset in both girls and boys. This advance in pubertal onset is not significant after adjusting for BMI, suggesting that higher BMI contributes to early pubertal onset (Sorensen et al, 2012). However, obesity is also associated with delayed puberty in boys (Crocker et al, 2014) and hypogonadotropic hypogonadism in adult males (Giagulli et al, 1994). In rare cases of congenital obesity caused by mutations in *LEP* and *LEPR*, it is also known that obesity is associated with GnRH deficiency. Thus the relationship between BMI and reproductive fitness is complex in humans. Further, obese individuals appear to have higher circulating FGF21 levels, consistent with a FGF21 resistant state (Fisher et al, 2010; Zhang et al, 2008). We have added this point in the Discussion (Page 19, Paragraph 2).

Minor comment:

Terminology: instead of mutants, they should call KLB-changes as variants. This study reports potential variants in CHH, but especially the potential bigenic roles remain to be verified.

We thank the Reviewer for this suggestion. We have changed the manuscript accordingly. In the revised manuscript, we used 'putative pathogenic variants' to define rare sequencing variants which are predicted to be deleterious. As is commonly practiced, we used the term mutation only after the variants were proven to be loss-of-function. For this reason, we changed the order of the paragraphs on the phenotype-genotype correlation and the functional assays in the Results.

References:

Avbelj Stefanija M, Jeanpierre M, Sykiotis GP, Young J, Quinton R, Abreu AP, Plummer L, Au MG, Balasubramanian R, Dwyer AA et al (2012) An ancient founder mutation in *PROKR2* impairs human reproduction. *Human molecular genetics* 21: 4314-4324

Boehm U, Bouloux PM, Dattani MT, de Roux N, Dode C, Dunkel L, Dwyer AA, Giacobini P, Hardelin JP, Juul A et al (2015) Expert consensus document: European Consensus Statement on congenital hypogonadotropic hypogonadism-pathogenesis, diagnosis and treatment. *Nature reviews Endocrinology* 11: 547-564

Crocker MK, Stern EA, Sedaka NM, Shomaker LB, Brady SM, Ali AH, Shawker TH, Hubbard VS, Yanovski JA (2014) Sexual dimorphisms in the associations of BMI and body fat with indices of pubertal development in girls and boys. *The Journal of clinical endocrinology and metabolism* 99: E1519-1529

de Roux N, Young J, Misrahi M, Genet R, Chanson P, Schaison G, Milgrom E (1997) A family with hypogonadotropic hypogonadism and mutations in the gonadotropin-releasing hormone receptor. *The New England journal of medicine* 337: 1597-1602

Fisher FM, Chui PC, Antonellis PJ, Bina HA, Kharitonov A, Flier JS, Maratos-Flier E (2010) Obesity is a fibroblast growth factor 21 (FGF21)-resistant state. *Diabetes* 59: 2781-2789

Fromer M, Moran JL, Chambert K, Banks E, Bergen SE, Ruderfer DM, Handsaker RE, McCarroll SA, O'Donovan MC, Owen MJ et al (2012) Discovery and statistical genotyping of copy-number variation from whole-exome sequencing depth. *American journal of human genetics* 91: 597-607

Giagulli VA, Kaufman JM, Vermeulen A (1994) Pathogenesis of the decreased androgen levels in obese men. *The Journal of clinical endocrinology and metabolism* 79: 997-1000

Goetz R, Ohnishi M, Ding X, Kurosu H, Wang L, Akiyoshi J, Ma J, Gai W, Sidis Y, Pitteloud N et al (2012) Klotho coreceptors inhibit signaling by paracrine fibroblast growth factor 8 subfamily ligands. *Molecular and cellular biology* 32: 1944-1954

Li Q, Wang K (2017) InterVar: Clinical Interpretation of Genetic Variants by the 2015 ACMG-AMP Guidelines. *American journal of human genetics* 100: 267-280

Richards S, Aziz N, Bale S, Bick D, Das S, Gastier-Foster J, Grody WW, Hegde M, Lyon E, Spector E et al (2015) Standards and guidelines for the interpretation of sequence variants: a joint consensus recommendation of the American College of Medical Genetics and Genomics and the Association for Molecular Pathology. *Genetics in medicine : official journal of the American College of Medical Genetics* 17: 405-424

Sorensen K, Mouritsen A, Aksglaede L, Hagen CP, Mogensen SS, Juul A (2012) Recent secular trends in pubertal timing: implications for evaluation and diagnosis of precocious puberty. *Horm Res Paediatr* 77: 137-145

Zhang X, Yeung DC, Karpisek M, Stejskal D, Zhou ZG, Liu F, Wong RL, Chow WS, Tso AW, Lam KS et al (2008) Serum FGF21 levels are increased in obesity and are independently associated with the metabolic syndrome in humans. *Diabetes* 57: 1246-1253

2nd Editorial Decision

13 June 2017

Thank you for the submission of your revised manuscript to EMBO Molecular Medicine. We have now received the enclosed reports from the referees that were asked to re-assess it. As you will see the reviewers are now globally supportive and I am pleased to inform you that we will be able to accept your manuscript pending the following final amendments:

- 1) Please address the minor comments of referee 1. Please provide a letter INCLUDING the reviewer's reports and your detailed responses to their comments (as Word file) and tune down the conclusion that are not fully supported by experimental data.
- 2) Please indicate in legends exact n= and exact p= values, not a range. Some people found that to keep the figures clear, providing a supplemental table with all exact p-values was preferable. You are welcome to do this if you want to.
- 3) Data deposition: we duly note that you did not obtain explicit consent to deposit the clinical data into a public repository. However I am afraid that you must do so. EGA for example allows access control of datasets should you need it. Please see below:

It is possible to submit information to the EGA while still continuing to manage access via a Data Access Committee (DAC): <https://www.ebi.ac.uk/ega/home>

It's important to stress that the Data Access Committee - which one would need to allow access to the raw data in some way - would remain unchanged. Many studies, each with managed access, do this (see: <https://www.ebi.ac.uk/ega/datasets>).

Be warned that it often takes quite a bit of time for submission. This can be fast tracked but more like 3 or 4 weeks rather than 3 or 4 days. This is because one's data access committee needs to be set up, documentation submitted around it, etc.

TEXT FROM EGA:

Who controls access to this dataset?

For each dataset that requires access control, there is a corresponding Data Access Committee (DAC) who determine access permissions. Data access requests are reviewed by the relevant DAC, not by the EGA.

The text within the study could look like this: "Our datasets were obtained from subjects who have

consented to the use of their individual genetic data for biomedical research, but not for unlimited public data release. Therefore, we submitted it to the European Genome-phenome Archive, through which researchers can apply for access of the raw data."

Please submit your revised manuscript within two weeks. I look forward to seeing a revised form of your manuscript as soon as possible.

***** Reviewer's comments *****

Referee #1 (Remarks):

In this revised manuscript Cheng Xu et al study the reproductive phenotype of the heterozygous KlbHet mice and now show that they exhibit disrupted estrous cycles, blunted LH levels at estrus stage and impaired fertility. Thus a mechanism of haploinsufficiency is feasible in the patients with heterozygous loss of function mutations of Klb.

However it seems that the number of litters/female mice is more important in homozygous KlbKO mice than in the case of heterozygous KlbHet mice. This is very unlikely. Can the authors check the figures (EV1 and Fig 3) and report complementary comparative studies? What is the reproductibility of such studies?

The authors try also to highlight a dominant negative mechanism by co-immunoprecipitation experiments. The experiments performed showed no difference in co-immunoprecipitation of FGFR1 when WT or mutants Klb were used. However this does not allow any conclusion on a possible dominant negative effect. Only a competitive inhibition of the formation of the Klb-FGFR1C-FGFR1 complex in the presence of increasing concentrations of mutated Klb would allow to highlight an inhibitory effect of the mutated Klb on the WT allele. Thus these conclusions must be deleted from the manuscript.

Referee #2 (Comments on Novelty/Model System):

The manuscript has improved considerably, and meets now high standard.

Referee #2 (Remarks):

The authors have satisfactorily responded to my criticism and I have no further comments. This is an interesting study.

2nd Revision - authors' response

23 June 2017

We thank you for your consideration of our manuscript. We have responded to Reviewers' comments and have adapted the manuscript accordingly. We have also made amendments according to your instructions, the major changes including:

- 1) We double checked the statistical analyses and made the figure legends more precise, with an addition of a supplementary table for all the exact *p* values.
- 2) The clinical and exome dataset deposition in European Genome-phenome Archive is ongoing. We have added the data deposition information in the Methods (Page 22, paragraph 1). We will inform you the accession number as soon as it is available.

Responses to Reviewers
Referee #1 (Remarks):

We thank the Reviewer for his positive comments on our revised manuscript.

In this revised manuscript Cheng Xu et al study the reproductive phenotype of the heterozygous KlbHet mice and now show that they exhibit disrupted estrous cycles, blunted LH levels at estrus stage and impaired fertility. Thus a mechanism of haploinsufficiency is feasible in the patients with heterozygous loss of function mutations of Klb.

However it seems that the number of litters/female mice is more important in homozygous KlbKO mice than in the case of heterozygous KlbHet mice. This is very unlikely. Can the authors check the figures (EVI and Fig 3) and report complementary comparative studies? What is the reproductibility of such studies?

As indicated in the Methods for the fertility assessment in mice, we performed a continuous mating protocol of 7 months in KlbKO mice. Due to the time limitation for the revision, we performed a short-term fertility test in KlbHET mice. The two protocols measure different parameters of fertility: the continuous protocol reports the number of litters per female and litter size, while the short-term protocol evaluates the rate of pregnancy. Therefore, the results cannot be used to compare the degree of a fertility defect between KlbKO and KlbHET. We have modified the Results, the Methods and the legend of Figure 4 (Page 13, paragraph 2; Page 25, paragraph 3; Page 48, paragraph 1) to further clarify the different tests in KlbKO and KlbHET mice.

The authors try also to highlight a dominant negative mechanism by co-immunoprecipitation experiments. The experiments performed showed no difference in co-immunoprecipitation of FGFR1 when WT or mutants Klb were used. However this does not allow any conclusion on a possible dominant negative effect. Only a competitive inhibition of the formation of the Klb-FGFR1c-FGFR1 complex in the presence of increasing concentrations of mutated Klb would allow to highlight an inhibitory effect of the mutated Klb on the WT allele. Thus these conclusions must be deleted from the manuscript.

Although our co-IP result is suggestive of a possible dominant negative effect, we agree with the Reviewer that only a competitive inhibition of receptor complex formation in the presence of increasing concentrations of KLB mutants could confirm the dominant negative effect. The co-IP assay still provides insights into the functional impact of the KLB mutant-FGFR1c complex on signaling transmission. We have thus modified the Results and the Discussion accordingly (Page 8, paragraph 3; Page 18, paragraph 1).

Referee #2 (Comments on Novelty/Model System):

The manuscript has improved considerably, and meets now high standard.

Referee #2 (Remarks):

The authors have satisfactorily responded to my criticism and I have no further comments. This is an interesting study.

We thank the Reviewer for his positive comments on our revised manuscript.

Corresponding Author Name: Nelly Pitteloud

Manuscript Number: EMM-2016-07376